# A microscopy-based kinetic analysis of yeast vacuolar protein sorting

**Jason C Casler, Benjamin S Glick\***

Department of Molecular Genetics and Cell Biology, University of Chicago, Chicago, United States

**Abstract** *Saccharomyces cerevisiae* is amenable to studying membrane traffic by live-cell fluorescence microscopy. We used this system to explore two aspects of cargo protein traffic through prevacuolar endosome (PVE) compartments to the vacuole. First, at what point during Golgi maturation does a biosynthetic vacuolar cargo depart from the maturing cisternae? To address this question, we modified a regulatable fluorescent secretory cargo by adding a vacuolar targeting signal. Traffic of the vacuolar cargo requires the GGA clathrin adaptors, which arrive during the early-to-late Golgi transition. Accordingly, the vacuolar cargo begins to exit the Golgi near the midpoint of maturation, significantly before exit of a secretory cargo. Second, how are cargoes delivered from PVE compartments to the vacuole? To address this question, we tracked biosynthetic and endocytic cargoes after they had accumulated in PVE compartments. The results suggest that stable PVE compartments repeatedly deliver material to the vacuole by a kiss-and-run mechanism.

## Introduction

Budding yeast has been instrumental for defining mechanisms of membrane traffic. Genetic screens of *Saccharomyces cerevisiae* have identified many conserved components of the biosynthetic and endocytic machineries (*Kaiser et al., 1997*; *Novick et al., 1980*). In addition, 4D (time-lapse 3D) fluorescence microscopy of *S. cerevisiae* has been powerful for characterizing membrane traffic pathways (*Day et al., 2016*; *Kurokawa et al., 2013*). Unlike most eukaryotes, *S. cerevisiae* has a non-stacked Golgi in which individual cisternae are optically resolvable by fluorescence microscopy (*Preuss et al., 1992*; *Wooding and Pelham, 1998*). This property enabled the first direct visualization of Golgi cisternal maturation (*Losev et al., 2006*; *Matsuura-Tokita et al., 2006*) as well as later studies of how maturation is regulated by GTPases and vesicle coat proteins (*Ishii et al., 2016*; *Kim et al., 2016*; *Papanikou et al., 2015*; *Rivera-Molina and Novick, 2009*; *Suda et al., 2013*; *Thomas and Fromme, 2020*).

Observations of the yeast Golgi can be synthesized in the following scheme (*Pantazopoulou and Glick, 2019*). New Golgi cisternae arise at ER exit sites and capture biosynthetic cargoes from the ER. These cisternae then mature by recycling resident Golgi proteins to the ER and to younger cisternae. During the early stage of maturation, one set of resident Golgi membrane proteins recycles with the aid of the COPI vesicle coat, whereas during the late stage of maturation, another set of resident Golgi membrane proteins recycles with the aid of the AP-1 clathrin adaptor. Biosynthetic cargoes are present in the cisternae throughout the maturation process (*Casler et al., 2019*; *Kurokawa et al., 2019*). Finally, the terminally mature Golgi cisternae fragment into secretory vesicles.

Recently, we expanded this analysis by examining the *S. cerevisiae* endocytic pathway. Our work was based on earlier studies of prevacuolar endosome (PVE) compartments, which are multivesicular bodies reminiscent of mammalian late endosomes (*Ma and Burd, 2020*; *Pelham, 2002*). The evidence indicates that *S. cerevisiae* has a minimal endomembrane system in which the late Golgi, also

**\*For correspondence:**
bsglick@uchicago.edu

**Competing interests:** The authors declare that no competing interests exist.

known as the *trans*-Golgi network (TGN), plays an additional role as an early and recycling endosome (*Day et al., 2018*). According to this view, yeast cells have two types of endosomes: (a) the late Golgi/TGN, and (b) PVE compartments that are typically attached to the vacuole.

4D microscopy can integrate these pictures of the biosynthetic and endocytic pathways by enhancing our understanding of cargo delivery to the vacuole. Such experiments require a way to visualize the transport of a biosynthetic vacuolar cargo in live yeast cells. To this end, we built on our recent engineering of a regulatable fluorescent secretory cargo (*Casler and Glick, 2019*; *Casler et al., 2019*). A tetrameric red fluorescent protein is fused to an improved dimerizing variant of the FK506-binding protein FKBP, and this construct is targeted to the ER lumen to generate aggregates, which are then solubilized with a ligand to create a fluorescent cargo wave that passes through the Golgi. We have now modified this construct by appending a tetrapeptide vacuolar targeting signal from the precursor to carboxypeptidase Y (CPY) (*Valls et al., 1990*). This targeting signal is recognized in the Golgi by the Vps10 cargo receptor (*Marcusson et al., 1994*), which in turn is packaged, with the aid of the GGA adaptors Gga1 and Gga2, into clathrin-coated vesicles destined for PVE compartments (reviewed in *Myers and Payne, 2013*). The result is that we have a regulatable fluorescent biosynthetic vacuolar cargo, which can be tracked as it moves from the ER through the Golgi to PVE compartments and then to the vacuole.

Our data extend prior results from other methods, which revealed the existence of traffic pathways for both biosynthetic and endocytic vacuolar cargoes. Biosynthetic vacuolar cargoes initially move from the ER to the Golgi. Some vacuolar membrane proteins traffic directly from the Golgi to the vacuole with the aid of the Golgi-associated AP-3 adaptor complex (*Cowles et al., 1997*; *Llinares et al., 2015*; *Odorizzi et al., 1998*). By contrast, CPY and certain other vacuolar hydrolases first traffic from the Golgi to PVE compartments (*Conibear and Stevens, 1998*; *Vida et al., 1993*). The same PVE compartments also contain endocytosed cargoes, such as the methionine permease Mup1, that are in transit to the vacuole (*Menant et al., 2006*). Within PVE compartments, transmembrane cargo proteins such as Mup1 are packaged into intraluminal vesicles that are transferred from the PVE compartments to the vacuole, where the intraluminal vesicles are degraded (*MacDonald et al., 2012*). Despite these insights, the understanding of vacuolar protein sorting remains incomplete. We have focused on two questions.

First, how is traffic from the Golgi to PVE compartments coordinated with cisternal maturation? The conventional view is that biosynthetic cargoes all travel together through the Golgi until reaching a terminal sorting stage (*De Matteis and Luini, 2008*; *Griffiths and Simons, 1986*), implying that vacuolar cargoes would remain in a cisterna throughout the maturation process. However, this idea is called into question by studies of GGA dynamics at the Golgi. We have now replicated work from the Payne lab showing that GGAs arrive earlier than AP-1 and at about the same time as Sec7 (*Daboussi et al., 2012*), an Arf guanine nucleotide exchange factor that is recruited and activated during the early-to-late Golgi transition (*Losev et al., 2006*; *McDonald and Fromme, 2014*). In addition, as described below, we have now tracked the passage of a fluorescent biosynthetic vacuolar cargo through the Golgi. The results indicate that the vacuolar cargo begins to depart when GGAs arrive, well before the final maturation of late Golgi cisternae into secretory vesicles. Thus, the late/TGN stage of Golgi maturation can be divided into a first sub-stage marked by the exit of vacuolar cargoes to PVE compartments, followed by a second sub-stage marked by AP-1-dependent intra-Golgi recycling and by the exit of secretory cargoes to the plasma membrane.

Second, how is material transferred from PVE compartments to the vacuole? Again, the answer was thought to be known, but 4D microscopy offers a new perspective. By analogy to mammalian cells, it was assumed that in yeast, early endosomes would mature into PVE compartments that would be consumed by fusing with the vacuole (*Balderhaar and Ungermann, 2013*; *Feyder et al., 2015*). Yet maturation of PVE compartments has not been observed—and indeed, if we are correct that yeast cells lack distinct early endosomes, then maturation of yeast endosomes is logically precluded (*Day et al., 2018*). The implication is that PVE compartments are not continually regenerated and must therefore be stable organelles. In support of this concept, our previous live-cell imaging revealed that PVE compartments undergo fission and homotypic fusion but otherwise persist indefinitely (*Day et al., 2018*). Evidence presented here suggests that stable PVE compartments deliver their contents to the vacuole by transient kiss-and-run fusion events.

# Results

## Addition of a tetrapeptide generates a fluorescent biosynthetic vacuolar cargo

To create a wave of fluorescent vacuolar cargo, we modified a recently developed regulatable fluorescent secretory cargo (*Casler and Glick, 2019*; *Casler et al., 2019*). Our secretory cargo consists of the tetrameric red fluorescent protein DsRed-Express2 fused to a dimerizing variant of FKBP, with a cleavable N-terminal signal sequence to direct cotranslational translocation into the ER. The signal sequence is followed by a tripeptide ER export signal (*Yin et al., 2018*) and a signal for *N*-linked glycosylation (*Figure 1B*). This fusion protein forms fluorescent aggregates within the ER lumen, and the aggregates can be dissolved by adding a synthetic ligand of FKBP (SLF) that blocks dimerization of FKBP (*Figure 1A*). Soluble tetramers then exit the ER in a nearly synchronized wave. Efficient

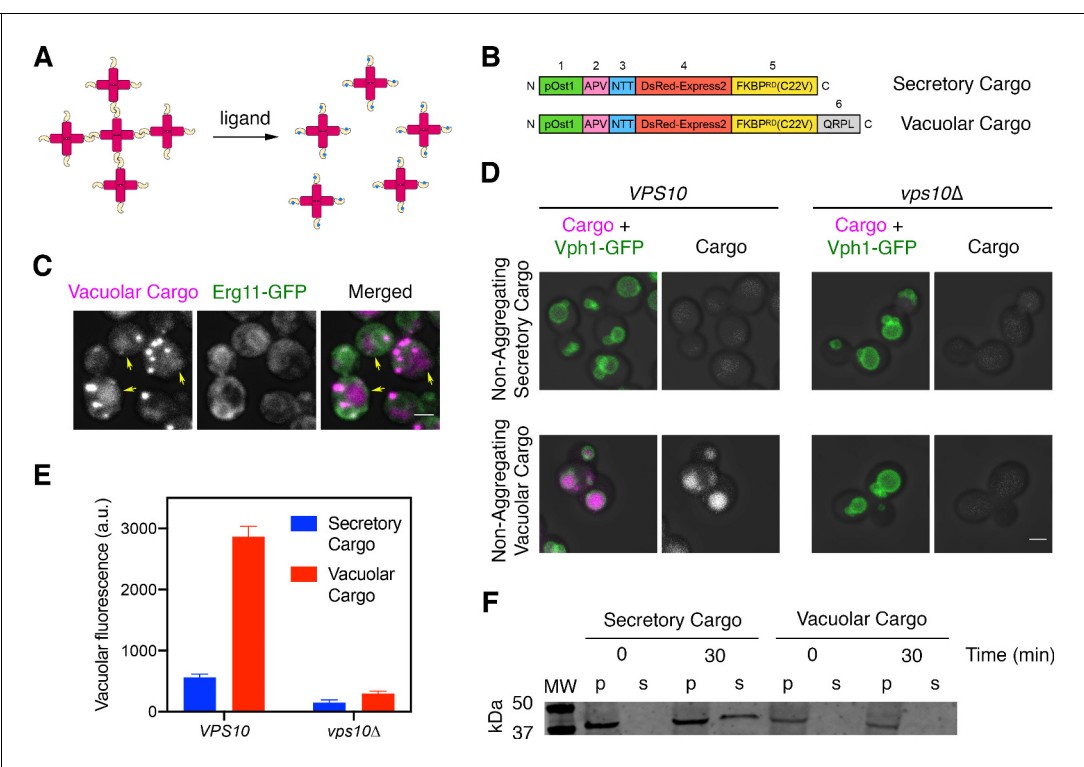

**Figure 1.** A regulatable vacuolar cargo. (A) General strategy for the use of reversibly aggregating fluorescent cargoes. DsRed-Express2 tetramers (red) are linked to a dimerizing FKBP variant (gold), so the tetramers associate to form aggregates. Addition of the FKBP ligand SLF (blue) blocks dimerization, thereby dissolving the aggregates into soluble tetramers that can exit the ER. (B) Functional regions of the reversibly aggregating secretory and vacuolar cargoes. The lengths of the regions are not to scale. 1: pOst1 (green) is an ER signal sequence that directs cotranslational translocation. 2: APV (pink) is a tripeptide signal for ER export. 3: NTT (blue) is a tripeptide signal for N-linked glycosylation. 4: DsRed-Express2 (red) is a tetrameric red fluorescent protein. 5: FKBP$^{RD}$(C22V) (gold) is a reversibly dimerizing variant of FKBP. 6: QRPL (gray) is a tetrapeptide signal for vacuolar targeting. (C) Aggregation in the ER of the vacuolar cargo. The ER membrane marker Erg11-GFP (green) confirms that the aggregates (magenta) are in the ER. Yellow arrows point to leaked cargo molecules that have accumulated in the vacuole. Shown are projected confocal Z-stacks. Scale bar, 2 μm. (D) Vacuolar targeting by the QRPL tetrapeptide. Non-aggregating variants of the secretory and vacuolar cargoes were expressed in *VPS10* wild-type or *vps10Δ* cells to visualize receptor-dependent targeting to the vacuole, which was marked by the vacuolar membrane marker Vph1-GFP. Significant vacuolar accumulation was seen only in the *VPS10* background when the QRPL signal was present. Shown are projected confocal Z-stacks. Scale bar, 2 μm. (E) Quantification of the cargo fluorescence signals in (D). The Vph1-GFP signal was used to create a mask for measuring cargo fluorescence in the vacuole. Data are average values from at least 69 cells for each strain. Fluorescence is plotted in arbitrary units (a.u.). Bars represent SEM. (F) Immunoblot to measure cell-associated and secreted levels of the secretory and vacuolar cargoes after SLF addition in rich medium. Cells expressing either the secretory or vacuolar cargo were grown to mid-log phase in YPD, washed with fresh YPD, and treated with SLF. At the 0 and 30 min time points, cell-associated pellet ('p') and secreted soluble ('s') fractions were separated by centrifugation. Samples were treated with endglycosidase H to trim *N*-linked glycans, and were analyzed by SDS-PAGE and immunoblotting. Shown is a representative example from four separate experiments. MW, molecular weight markers. The predicted molecular weights for the mature cargoes are ~38–39 kDa. In some samples, the cell-associated vacuolar cargo at the 30 min time point showed evidence of degradation, presumably due to exposure to vacuolar proteases (data not shown).

dissolution of the aggregates requires a drug-sensitive yeast strain. Thus, all of our experiments with the regulatable cargoes employed yeast strains containing deletions of the transcription factors Pdr1 and Pdr3, which mediate pleiotropic drug resistance (*Barrero et al., 2016*; *Casler et al., 2019*; *Coorey et al., 2015*; *Schüller et al., 2007*). We found previously that the regulatable secretory cargo persists in cisternae through the early-to-late transition of Golgi maturation, and that a fraction of the cargo molecules are recycled within the Golgi in an AP-1-dependent manner (*Casler et al., 2019*). The goal was to perform similar experiments with a modified cargo that was targeted to the vacuole.

Our strategy was to augment the fusion protein with a vacuolar targeting signal. An obvious candidate for this signal was the propeptide of the vacuolar hydrolase CPY. Prior studies showed that within the propeptide, the tetrapeptide QRPL is necessary and sufficient to direct CPY from the Golgi to PVE compartments by means of the sorting receptor Vps10 (*Johnson et al., 1987*; *Marcusson et al., 1994*; *Valls et al., 1987*; *Valls et al., 1990*). We flanked QRPL with glycine/serine spacers by appending at the C-terminus of the fusion protein the peptide GSQRPLGGS (*Figure 1B*). The C-terminus was chosen because insertion of QRPL near the N-terminus of the mature protein prevented robust ER aggregation (data not shown). Addition of the QRPL-containing peptide at the C-terminus preserved the formation of aggregates within the lumen of the ER, which was marked by GFP-tagged Erg11 (*Figure 1C*). Some cargo molecules were already present in the vacuole prior to dissolution of the aggregates (arrows in *Figure 1C*), presumably because signal-dependent ER exit allowed a fraction of the cargo molecules to escape from the ER and reach the vacuole while others became trapped in ER-localized aggregates (*Casler et al., 2019*). Based on these observations, the QRPL-containing construct was a candidate for a regulatable fluorescent vacuolar cargo.

To test if the QRPL signal worked as intended, we tested non-aggregating (and therefore non-regulatable) versions of the secretory and vacuolar cargoes. The vacuolar membrane was visualized with Vph1-GFP. Compared to the non-aggregating secretory cargo, which accumulated at low levels in the vacuole (*Casler et al., 2019*), the non-aggregating vacuolar cargo accumulated at high levels in the vacuole in a Vps10-dependent manner (*Figure 1D,E*). A final control experiment employed the regulatable versions of the cargoes once again. At 30 min after addition of SLF, the regulatable secretory cargo was detected in the culture medium whereas the regulatable vacuolar cargo was not (*Figure 1F*). These results confirm that the vacuolar cargo traffics efficiently to the vacuole. For convenience, from now on we will refer to the regulatable fluorescent secretory cargo and the regulatable fluorescent vacuolar cargo as the secretory and vacuolar cargoes, respectively.

## Traffic of the vacuolar cargo can be visualized

We first measured the overall rate of cargo traffic from the ER to the vacuole. A *VPS10* wild-type strain and a *vps10Δ* mutant strain expressed the vacuolar cargo together with the vacuole marker Vph1-GFP. After SLF was added to initiate cargo transport, the cells were imaged by 4D confocal microscopy for 60 min (*Figure 2—video 1* and *Figure 2A–C*). With *VPS10* cells, we saw a gradual accumulation of fluorescence in the vacuole. With *vps10Δ* cells, virtually no fluorescence appeared in the vacuole, presumably because the cargo exited the cell in secretory vesicles (see below, *Figure 6—figure supplement 2*). In typical *VPS10* cells, small amounts of the cargo were detected in the vacuole within 8–15 min after SLF addition, and full delivery to the vacuole required at least 40 min (*Figure 2B,C*). Individual cells showed significant variations in the timing of cargo traffic. About 35% of the cells required more than 15 min—and in some cases, more than 25 min—before any cargo appeared in the vacuole (*Figure 2B,D*). On average, the cell population showed a gradual increase in vacuolar fluorescence over a time course of an hour (*Figure 2C*).

A potential concern with this analysis is that during the time interval examined, new cargo molecules were being synthesized, and some of those molecules could have become fluorescent and reached the vacuole. We addressed this issue by repeating the experiment after pre-treating the cells with cycloheximide to block protein synthesis. Following SLF addition, untreated and cycloheximide-treated cells showed similar traffic kinetics, with cycloheximide causing only a modest reduction in the amount of cargo accumulating in the vacuole even though cell growth was arrested (*Figure 2—figure supplement 1*). We conclude that to a close approximation, the kinetics observed in the absence of cycloheximide reflect traffic of the vacuolar cargo molecules that were originally in ER-localized aggregates.

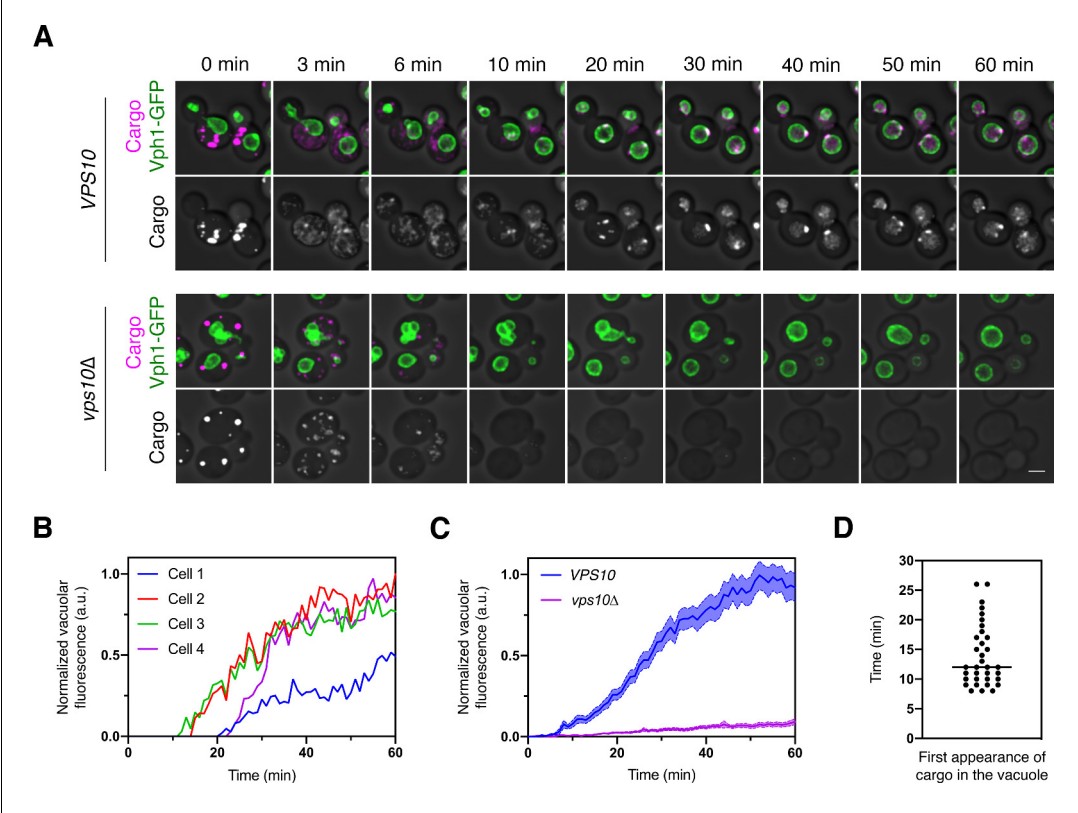

**Figure 2.** Traffic kinetics of the vacuolar cargo. (**A**) Visualizing cargo traffic. The vacuolar cargo expressed in *VPS10* wild-type or *vps10Δ* strains was imaged by 4D confocal microscopy. Prior to the video, fluorescence from leaked cargo molecules was bleached by illuminating the vacuole with a 561 nm laser at maximum intensity for 20–30 s. Then SLF was added, and Z-stacks were captured every minute for 60 min. The top panel shows the cargo (magenta) together with the vacuolar membrane marker Vph1-GFP (green), while the bottom panel shows only the cargo. Fluorescence data are superimposed on brightfield images of the cells. Shown are representative frames from *Figure 2—video 1*. Scale bar, 2 μm. (**B**) Quantification of the vacuolar fluorescence from each of the four *VPS10* cells in (**A**). The Vph1-GFP signal was used to create a mask for measuring cargo fluorescence in the vacuole. Fluorescence is plotted in arbitrary units (a.u.). (**C**) Quantification of the average vacuolar fluorescence in *VPS10* and *vps10Δ* cells after addition of SLF. For each strain, at least 39 cells were analyzed from four movies. Quantification was performed as in (**B**). The shaded borders represent SEM. (**D**) Quantification of the first appearance of cargo fluorescence in the vacuole. Data are from the same set of *VPS10* cells analyzed for (**C**). Appearance in the vacuole was scored as the first time point at which the vacuolar cargo fluorescence reached at least 5% of its final value.

The online version of this article includes the following video and figure supplement(s) for figure 2:

**Figure supplement 1.** Minor effect of cycloheximide treatment on the kinetics of cargo traffic to the vacuole.

**Figure 2—video 1.** Visualizing traffic of the vacuolar cargo from the ER to the vacuole.

https://elifesciences.org/articles/56844#fig2video1

Why is traffic to the vacuole so slow? For comparison, secretory cargo molecules can travel from the ER to the plasma membrane within 5–10 min, and nearly all of them are secreted within 20 min (*Casler et al., 2019*; *Losev et al., 2006*). This effect is seen in the *vps10Δ* cells because in the absence of a sorting receptor, the vacuolar cargo behaves like a secretory cargo (*Figure 2A*). To understand the slow kinetics of cargo delivery to the vacuole, we set out to track the different steps of this pathway by fluorescence microscopy.

## The vacuolar cargo transits rapidly through the Golgi and accumulates in PVE compartments

Early work suggested that the rate-limiting step in biosynthetic cargo transport to the vacuole is exit from PVE compartments (*Vida et al., 1993*). To test this idea, we used 4D confocal movies to visualize the vacuolar cargo together with organellar markers. PVE compartments were labeled by tagging Vps8, a subunit of the CORVET tethering complex (*Arlt et al., 2015*; *Markgraf et al., 2009*). We showed previously that tagged Vps8 colocalized strongly with a variety of other PVE compartment

markers, and that the observed dynamics of PVE compartments were similar when using either tagged Vps8 or other markers (*Day et al., 2018*). For three-color 4D movies, the red fluorescent vacuolar cargo was visualized together with the early Golgi marker GFP-Vrg4 and the late Golgi marker Sec7-HaloTag, or together with the PVE marker Vps8-GFP and the vacuole marker Vph1-HaloTag (*Arlt et al., 2015*; *Day et al., 2018*; *Losev et al., 2006*). In this and subsequent experiments, Halo-Tag was conjugated to the far-red dye JF$_{646}$ (*Grimm et al., 2015*). After the cargo aggregates were solubilized with SLF, a strain expressing the vacuolar cargo plus the Golgi markers was imaged every 30 s for 29.5 min, and a strain expressing the vacuolar cargo plus the PVE and vacuole markers was imaged every 60 s for 60 min. The results showed cargo accumulation within Golgi compartments 1–5 min after SLF addition, followed by nearly complete transfer of the cargo to PVE compartments by 10 min (*Figure 3—video 1* and *Figure 3—video 2* and *Figure 3A–D*). After 10 min, puncta that contained the cargo invariably labeled with Vps8-GFP, confirming that Vps8 is a reliable marker for PVE compartments. The cargo gradually exited the PVE compartments and then accumulated in the vacuole as described above. Interestingly, even though the PVE compartments contained cargo by 10 min, some of them did not immediately begin to transfer cargo to the vacuole (see *Figure 2B,D*), suggesting that PVE compartments can be temporarily quiescent with regard to cargo delivery. These results verify that the rate-limiting step in traffic to the vacuole is not movement through the Golgi, but rather transfer of the cargo from PVE compartments to the vacuole.

Interestingly, although most of the early Golgi cisternae contained detectable vacuolar cargo at early time points, only about half of the late Golgi cisternae ever contained detectable vacuolar cargo (*Figure 3C*). By contrast, we previously saw that the fluorescent secretory cargo was present in nearly all of the late Golgi cisternae, where it persisted until the final phase of maturation (*Casler et al., 2019*). A possible explanation is that the vacuolar cargo departed during the late stage of Golgi maturation, so that as late Golgi cisternae became more mature, they no longer contained fluorescent cargo. To test this hypothesis, we next visualized the dynamics of the vacuolar cargo in maturing Golgi cisternae.

## The vacuolar cargo begins to exit the Golgi near the midpoint of cisternal maturation

To determine when the vacuolar cargo departs from maturing cisternae, we performed 4D confocal microscopy of yeast cells expressing the vacuolar cargo, the early Golgi marker GFP-Vrg4, and the late Golgi marker Sec7-HaloTag. As previously described, we readily detected Golgi maturation events in which GFP-Vrg4-labeled cisternae lost the GFP-Vrg4 signal as they acquired Sec7-HaloTag, which they subsequently lost in the final phase of maturation (*Casler et al., 2019*; *Losev et al., 2006*). Intriguingly, the vacuolar cargo signal always began to decline near the midpoint of maturation (*Figure 4—video 1*, *Figure 4A,B*, and *Figure 4—figure supplement 1A–C*). The rate of decline varied between cells, and qualitative observations indicated that the decline was slower in cells expressing very high levels of the vacuolar cargo (data not shown), probably because the sorting machinery was saturated. Therefore, we focused the analysis on cells expressing moderate levels of the vacuolar cargo. The average behavior from 21 maturation events is depicted in *Figure 4C*.

We predicted that removal of the Vps10 sorting receptor would prevent normal exit of the vacuolar cargo, which would then behave like a secretory cargo. This prediction was tested by tracking the vacuolar cargo during Golgi maturation in a *vps10Δ* mutant. In the absence of Vps10, the vacuolar cargo signal no longer declined during the early-to-late transition (*Figure 4—video 2*, *Figure 4D–F*, and *Figure 4—figure supplement 1D–F*). Instead, *vps10Δ* cells actually displayed a transient increase in the cargo signal during the early-to-late transition, likely due to AP-1-dependent recycling from older cisternae as previously shown for the fluorescent secretory cargo (*Casler et al., 2019*). These data indicate that in wild-type cells, the vacuolar cargo begins to exit the Golgi in a Vps10-dependent manner around the time of the early-to-late transition.

## GGAs but not AP-1 are required to sort the vacuolar cargo

To characterize how the vacuolar cargo exits the Golgi, we tested the roles of the AP-1 clathrin adaptor and of the GGA clathrin adaptors Gga1 and Gga2. AP-1 was originally proposed to mediate transport of proteins from the TGN to endosomes (reviewed in *Hinners and Tooze, 2003*), but subsequent work implicated GGAs in TGN-to-endosome traffic in both yeast and mammalian cells

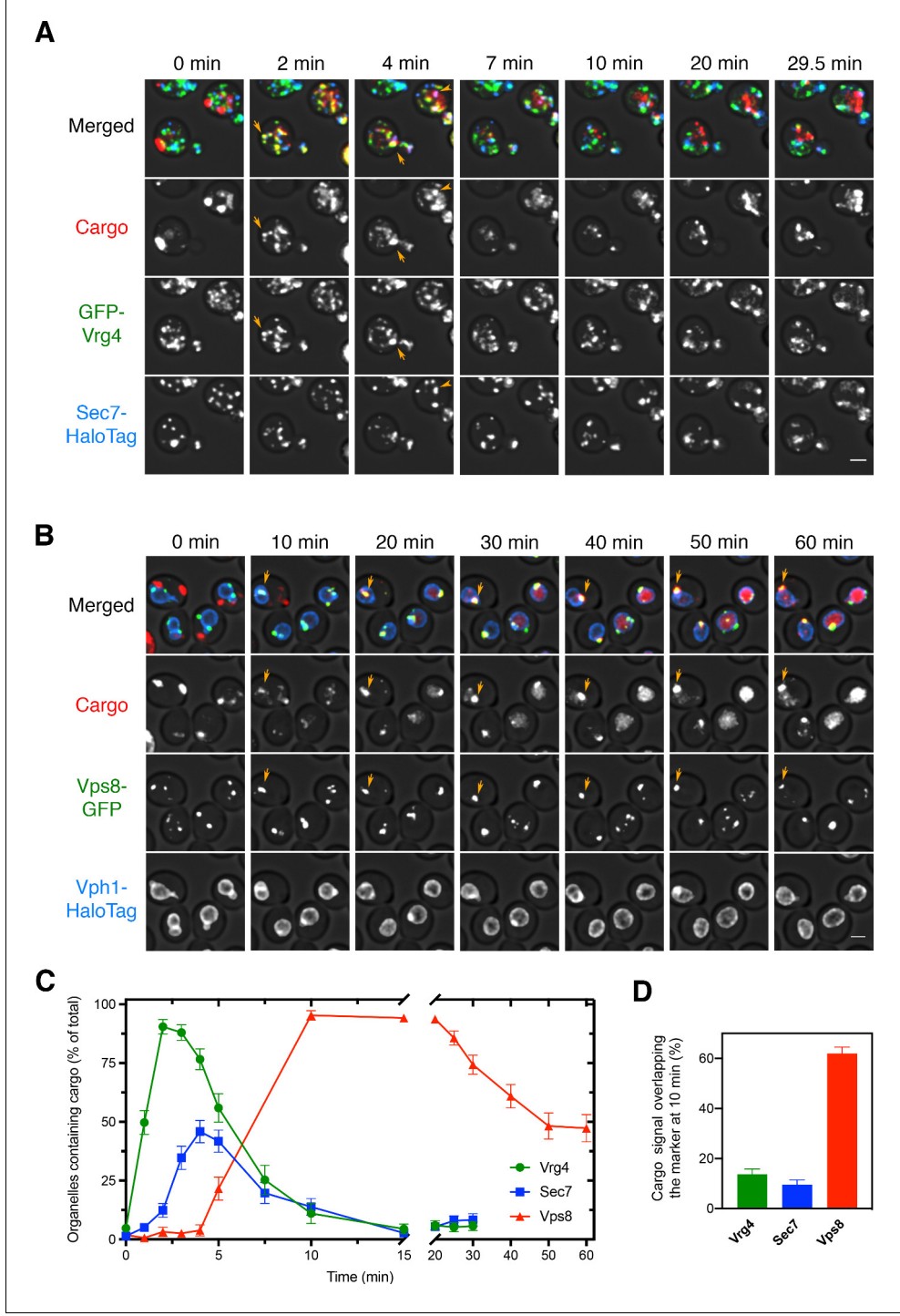

**Figure 3.** Sequential appearance of the vacuolar cargo in Golgi and PVE compartments. (**A**) Appearance of the vacuolar cargo in early Golgi compartments marked with GFP-Vrg4 and in late Golgi compartments marked with Sec7-HaloTag. Cells were grown to mid-log phase, labeled with JF$_{646}$, and imaged by 4D confocal microscopy. Prior to beginning the video, fluorescence from leaked cargo molecules in the vacuole was bleached by illuminating with maximum intensity 561 nm laser power for 20–30 s. SLF was added directly to the dish between the first and second Z-stacks, and then additional Z-stacks were captured every 30 s for 29.5 min. Images are representative time points from *Figure 3—video 1*. The top panel shows the merged images, and the other panels show the individual fluorescence channels for cargo, Vrg4, and Sec7. Scale bar, 2 μm. (**B**) Appearance of the vacuolar cargo in PVE compartments marked with Vps8-GFP and in the vacuole marked with Vph1-HaloTag.
*Figure 3 continued on next page*

*Figure 3 continued*

The procedure was as in (**A**), except that Z-stacks were captured every 60 s for 60 min. Images are representative time points from *Figure 3—video 2*. The top panel shows the merged images, and the other panels show the individual fluorescence channels for cargo, Vps8, and Vph1. Scale bar 2 µm. (**C**) Quantification of the percentage of compartments containing detectable cargo from (**A**) and (**B**). Confocal movies were average projected and manually scored for the presence of cargo in labeled compartments. For each strain, at least 26 cells were analyzed from four movies. The bars represent SEM. (**D**) Quantification of the percentage of the total cargo fluorescence present in early Golgi, late Golgi, and PVE compartments 10 min after SLF addition. The fluorescence for a compartment marker was used to generate a mask to quantify the corresponding cargo fluorescence. Data were taken from at least 26 cells from four movies. The bars represent SEM.

The online version of this article includes the following video(s) for figure 3:

**Figure 3—video 1.** Visualizing traffic of the vacuolar cargo together with Golgi markers.
https://elifesciences.org/articles/56844#fig3video1
**Figure 3—video 2.** Visualizing traffic of the vacuolar cargo together with PVE and vacuole markers.
https://elifesciences.org/articles/56844#fig3video2

---

(*Black and Pelham, 2000*; *Dell'Angelica et al., 2000*; *Hirst et al., 2000*; *Zhdankina et al., 2001*). In *S. cerevisiae*, AP-1 localizes exclusively to the late Golgi, and it mediates intra-Golgi recycling of some resident late Golgi proteins and secretory cargoes (*Casler and Glick, 2019*; *Day et al., 2018*; *Liu et al., 2008*; *Papanikou et al., 2015*; *Spang, 2015*; *Valdivia et al., 2002*). It was previously reported that yeast GGAs function upstream of AP-1 and that GGAs display similar kinetics of arrival and departure as the late Golgi reference marker Sec7 (*Daboussi et al., 2012*). However, a somewhat different conclusion was presented in a more recent study, which reported that GGAs arrived at Golgi cisternae significantly later than Sec7 (*Tojima et al., 2019*). To clarify the functions of these adaptors in sorting the vacuolar cargo, we combined a kinetic analysis of AP-1 and GGA dynamics with tests of deletion mutants.

If a given adaptor is involved in transporting the vacuolar cargo out of the Golgi, then arrival of that adaptor is expected to coincide with initiation of cargo departure. In a kinetic analysis, we first compared Sec7 with the AP-1 subunit Apl2 and with the major GGA isoform Gga2 (*Myers and Payne, 2013*). Three-color imaging was performed with Apl2-GFP, Gga2-HaloTag, and Sec7-mScarlet. Gga2 consistently showed arrival and departure kinetics nearly identical to those of Sec7, whereas Apl2 consistently arrived ~20–40 s later and departed ~10–15 s later than Gga2 and Sec7 (*Figure 5—video 1*, *Figure 5A,B*, and *Figure 5—figure supplement 1A–C*). Thus, GGAs arrive at the Golgi at about the same time that the vacuolar cargo begins to depart. Indeed, when the vacuolar cargo was visualized in maturing cisternae together with the early Golgi marker GFP-Vrg4 and with Gga2-HaloTag, the first appearance of Gga2 occurred at about the same time that the vacuolar cargo signal started to decline (*Figure 5—video 2*, *Figure 5C,D*, and *Figure 5—figure supplement 1D*). Interestingly, the cargo signal sometimes began to drop a short time before Gga2 was visible at the Golgi. A potential explanation is that when GGAs are first recruited, they are immediately packaged into vesicles that transport vacuolar cargoes from the Golgi, so the pool of Golgi-associated Gga2 initially remains low. Regardless of whether this interpretation is correct, GGA arrival kinetics closely match the departure kinetics of the vacuolar cargo, whereas AP-1 arrival kinetics do not.

If a given adaptor is involved in transporting the vacuolar cargo out of the Golgi, then loss of that adaptor should disrupt normal sorting. We generated an *apl4Δ* deletion strain to inactivate AP-1 and a *gga1Δ gga2Δ* double deletion strain to inactivate GGAs, and then tracked the vacuolar cargo together with the early Golgi marker GFP-Vrg4 and the late Golgi marker Sec7-HaloTag. In the *apl4Δ* strain, the vacuolar cargo signal began to decline during the early-to-late transition of Golgi maturation in a manner indistinguishable from that seen in wild-type cells (*Figure 6—video 1*, *Figure 6A–C*, and *Figure 6—figure supplement 1A–C*). Strikingly, in the *gga1Δ gga2Δ* strain, the vacuolar cargo persisted during the early-to-late transition (*Figure 6—video 2*, *Figure 6D–F*, and *Figure 6—figure supplement 1D-F*). In most of the events analyzed for the *gga1Δ gga2Δ* strain, the vacuolar cargo could be detected within the cisterna until the Sec7 signal disappeared or even afterwards (*Figure 6D,E* and *Figure 6—figure supplement 1D,E*). A caveat is that the *gga1Δ gga2Δ* strain displayed somewhat altered Vrg4 and Sec7 maturation kinetics, as indicated by abnormally

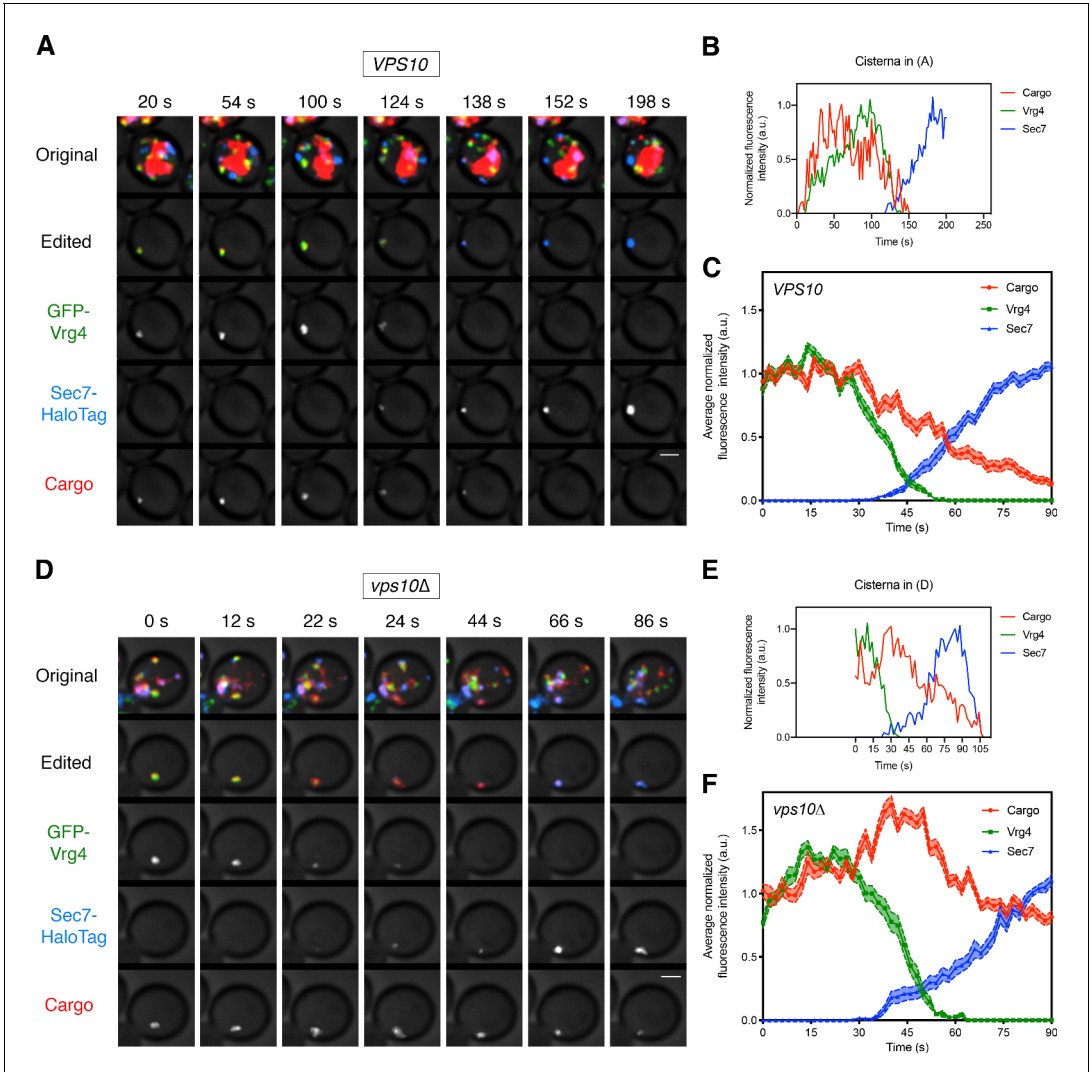

**Figure 4.** Visualizing the vacuolar cargo during Golgi maturation. (**A**) Visualizing the vacuolar cargo in a *VPS10* wild-type strain. Cells expressing the vacuolar cargo together with the early Golgi marker GFP-Vrg4 and the late Golgi marker Sec7-HaloTag were grown to mid-log phase, labeled with JF$_{646}$, and imaged by 4D confocal microscopy. SLF was added 1–3 min before imaging. Shown are average projected Z-stacks at representative time points from *Figure 4—video 1*. The top row shows the complete projection, the second row shows an edited projection that includes only the cisterna being tracked, and the other rows show the individual fluorescence channels from the edited projection. The large red structure is the vacuole, which contained cargo molecules that had escaped from the ER prior to SLF addition as described in *Figure 1*. Scale bar, 2 μm. (**B**) Quantification of the fluorescence intensities of the Golgi markers and the vacuolar cargo during a typical maturation event. Depicted are the normalized fluorescence intensities in arbitrary units (a.u.) of the cisterna tracked in (**A**). (**C**) Average cargo signal during the early-to-late Golgi transition. For 21 maturation events from 18 movies of cells expressing moderate levels of the vacuolar cargo, fluorescence was quantified over a 90 s window with Z-stacks collected every 2 s. Normalization was performed by defining the maximum value as the average of the first six fluorescence values for the cargo and Vrg4, or of the last six fluorescence values for Sec7. Traces were aligned at the midpoint of the Vrg4-to-Sec7 transition, and the normalized fluorescence signals were averaged. The shaded borders represent SEM. (**D**) Visualizing the vacuolar cargo in a *vps10Δ* strain. The experiment was performed as in (**A**). Shown are average projected Z-stacks at representative time points from *Figure 4—video 2*. (**E**) Quantification of the fluorescence intensities of the Golgi markers and the vacuolar cargo during a typical maturation event in the *vps10Δ* strain. Depicted are the normalized fluorescence intensities in arbitrary units (a.u.) of the cisterna tracked in (**D**). (**F**) Average cargo signal during the early-to-late Golgi transition in a *vps10Δ* strain. The experiment was performed as in (**C**). Data were collected for 12 maturation events from 12 movies of cells expressing moderate levels of the vacuolar cargo.

The online version of this article includes the following video and figure supplement(s) for figure 4:

**Figure supplement 1.** Additional examples of vacuolar cargo traffic during Golgi maturation.

**Figure 4—video 1.** Visualizing traffic of the vacuolar cargo during Golgi maturation.

https://elifesciences.org/articles/56844#fig4video1

**Figure 4—video 2.** Visualizing traffic of the vacuolar cargo during Golgi maturation in a strain lacking Vps10.

*Figure 4 continued on next page*

*Figure 4 continued*

https://elifesciences.org/articles/56844#fig4video2

shallow slopes for the arrival and departure curves of the Golgi markers (*Figure 6F*). Nevertheless, maturation events could be readily identified in the *gga1Δ gga2Δ* strain, and the results strongly suggest that GGAs are needed for the vacuolar cargo to exit the Golgi.

As a further test of this interpretation, we hypothesized that loss of GGAs would cause the vacuolar cargo to be secreted, as is true for native CPY (*Dell'Angelica et al., 2000*; *Hirst et al., 2000*; *Zhdankina et al., 2001*). This prediction was tested by an immunoblot of media samples collected 30 min after solubilizing the vacuolar cargo with SLF in wild-type, *vps10Δ*, *apl4Δ*, and *gga1Δ gga2Δ* strains. Secreted cargo was consistently observed with the *vps10Δ* and *gga1Δ gga2Δ* strains but not with the wild-type or *apl4Δ* strains (*Figure 6—figure supplement 2A*). As another readout for mistargeting of the vacuolar cargo, we compared the total amount of cargo present in the vacuole 60 min after solubilizing the cargo with SLF in wild-type, *vps10Δ*, *apm3Δ*, *apl4Δ*, and *gga1Δgga2Δ* strains. Apm3 serves as an additional control because it is a subunit of the AP-3 adaptor complex, which functions in a parallel pathway that targets certain membrane proteins directly from the Golgi to the vacuole (*Myers and Payne, 2013*; *Odorizzi et al., 1998*). The wild-type, *apl4Δ*, and *apm3Δ* strains all showed similar amounts of vacuolar cargo, while the *vps10Δ* and *gga1Δ gga2Δ* strains showed very little vacuolar cargo (*Figure 6—figure supplement 2B,C*). The combined results indicate that the vacuolar cargo is sorted to PVE compartments after the early-to-late Golgi transition, with the aid of Vps10 and GGAs but with no significant contribution from AP-1.

## Soluble and transmembrane cargoes apparently transit from PVE compartments to the vacuole by kiss-and-run events

Based on the observation that PVE compartments are indefinitely long-lived, we have speculated that PVE compartments deliver cargoes to the vacuole via kiss-and-run fusion events that generate transient pores (*Day et al., 2018*). Such pores would allow the passage of soluble cargoes as well as intraluminal vesicles that carry membrane-bound cargoes (*Henne et al., 2011*). This type of partial fusion mechanism could explain why the movement of cargoes from PVE compartments to the vacuole is relatively slow.

To test this concept, we developed an assay in which a pool of labeled cargo molecules in a PVE compartment can be tracked during passage to the vacuole. This approach had two technical hurdles: (1) prior to imaging, a fraction of the fluorescent cargo molecules are in the vacuole or other non-PVE compartments, and (2) a typical cell contains several PVE compartments that can undergo fission and homotypic fusion (*Day et al., 2018*), making it difficult to follow individual organelles. The problem with background fluorescence was addressed by photobleaching cargo molecules outside the PVE compartments. Bleaching occurred in the interior of the vacuole and in the portion of the cell outside the vacuole, leaving a ring of non-bleached fluorescence that included vacuole-associated PVE compartments. The problem of tracking multiple PVE compartments was addressed by focusing on cells that happened to contain just one or two punctate PVE structures.

Traffic of the biosynthetic vacuolar cargo from PVE compartments to the vacuole was visualized by generating a yeast strain expressing the vacuolar cargo together with the PVE marker Vps8-GFP and the vacuole marker Vph1-HaloTag. At 10–15 min after SLF addition, the majority of the cargo molecules had passed through the Golgi and accumulated in PVE compartments (see *Figure 3C* above). 4D confocal imaging was then performed after photobleaching the signal outside the PVE compartments. As previously documented (*Day et al., 2018*), the Vps8-GFP-labeled PVE compartments were persistent structures. Meanwhile, cargo fluorescence moved from PVE compartments to the vacuole. This process was examined by capturing 10 min movies. Some PVE compartments were quiescent for cargo delivery as described above, so the analysis focused on PVE compartments that were active throughout the imaging period. During the majority of a typical time course, the fluorescence in a PVE compartment gradually declined, and this decrease was matched by a gradual increase of fluorescence in the vacuole (*Figure 7—video 1* and *Figure 7A,B*). Occasionally, a discrete event rapidly transferred a substantial fraction of the cargo fluorescence to the vacuole (*Figure 7—video 2* and *Figure 7C,D*). Although the Vps8-GFP signal varied, and sometimes dropped

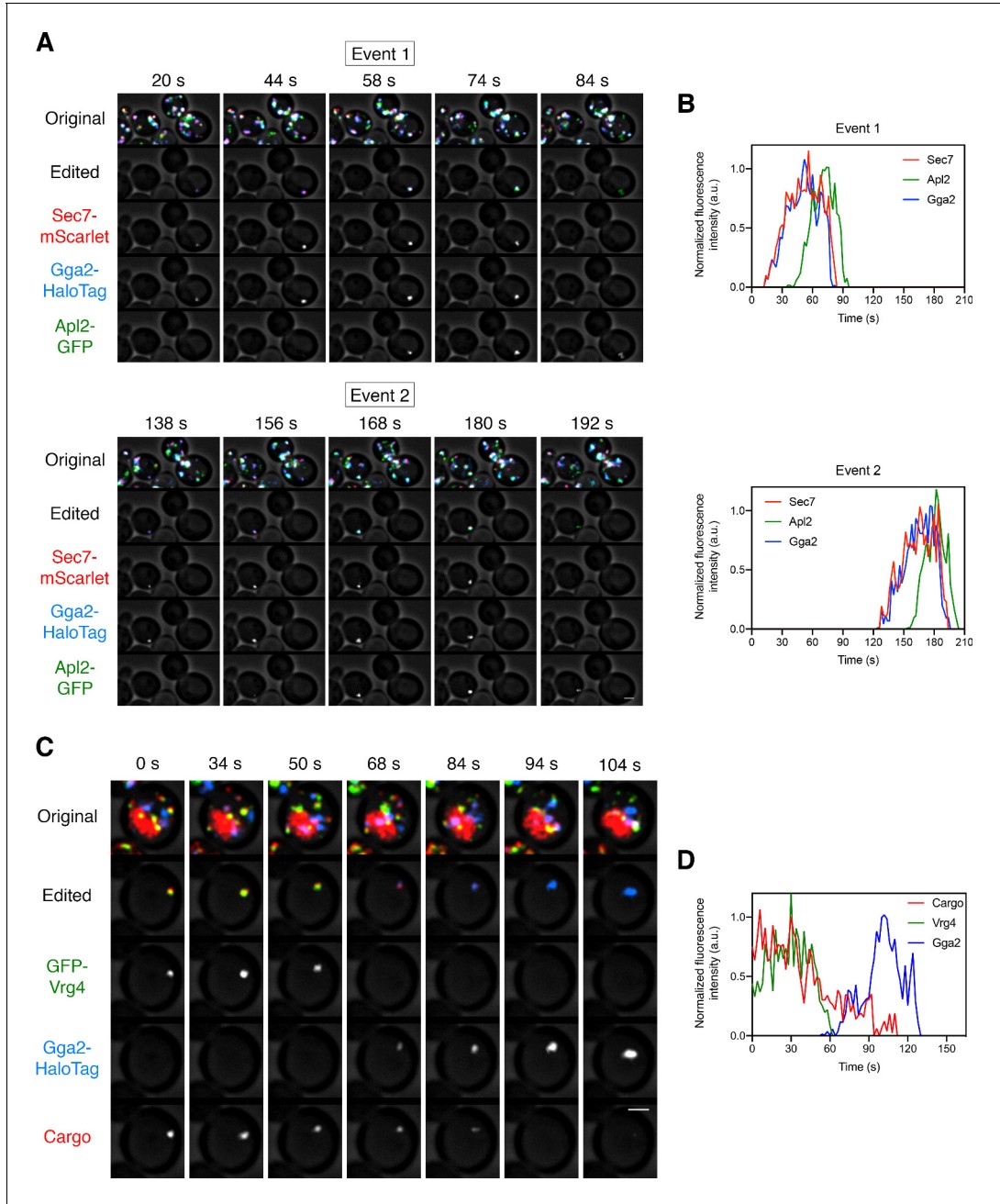

**Figure 5.** Kinetics of GGA arrival at the Golgi. (**A**) Visualizing the dynamics of the GGA and AP-1 adaptors during cisternal maturation. A strain expressing the GGA protein Gga2-HaloTag, the AP-1 subunit Apl2-GFP, and the late Golgi marker Sec7-mScarlet was grown to mid-log phase, labeled with JF$_{646}$, and imaged by 4D confocal microscopy. Shown are average projected Z-stacks at representative time points from *Figure 5—video 1*. The top row shows the complete projection, the second row shows an edited projection that includes only the cisterna being tracked, and the other rows show the individual fluorescence channels from the edited projection. Two maturation events are highlighted. Scale bar, 2 μm. (**B**) Quantification of the fluorescence intensities of the late Golgi marker and the adaptors during typical maturation events. Depicted are the normalized fluorescence intensities in arbitrary units (a.u.) of the two cisternae tracked in (**A**). (**C**) Visualizing Vrg4 and Gga2 together with the vacuolar cargo. The experiment was performed as in *Figure 4A*, except that the Golgi markers were GFP-Vrg4 and Gga2-HaloTag. Shown are average projected Z-stacks at representative time points from *Figure 5—video 2*. The top row shows the complete projection, the second row shows an edited projection that includes only the cisterna being tracked, and the other rows show the individual fluorescence channels from the edited projection. Scale bar, 2 μm. (**D**) Quantification of the fluorescence intensities of the Golgi markers together with the vacuolar cargo during a typical maturation event. Depicted are the normalized fluorescence intensities in arbitrary units (a.u.) of the cisterna tracked in (**C**).

The online version of this article includes the following video and figure supplement(s) for figure 5:

**Figure supplement 1.** Additional examples of adaptor dynamics and of the relationship between GGA arrival and vacuolar cargo departure.

*Figure 5 continued on next page*

*Figure 5 continued*

**Figure 5—video 1.** Visualizing the dynamics of the GGA and AP-1 clathrin adaptors during Golgi maturation.

https://elifesciences.org/articles/56844#fig5video1

**Figure 5—video 2.** Visualizing the vacuolar cargo together with a GGA adaptor.

https://elifesciences.org/articles/56844#fig5video2

after a discrete event, the punctate PVE compartments marked with Vps8-GFP were never seen to disappear. Our interpretation is that the vacuolar cargo moves from long-lived PVE compartments to the vacuole by a series of kiss-and-run events that are frequently small and sometimes large.

We wondered whether the putative kiss-and-run events could also deliver transmembrane cargoes that are encapsulated in intraluminal vesicles within PVE compartments (*Henne et al., 2011*). This experiment employed the methionine permease Mup1 as a model transmembrane cargo (*Menant et al., 2006*). In the absence of methionine in the culture medium, Mup1 resides in the plasma membrane, and upon addition of methionine, Mup1 is ubiquitinated, internalized, and sent to the vacuole for degradation (*Lin et al., 2008*). We generated a strain expressing Mup1-mScarlet, together with Vps8-GFP to label PVE compartments and Vph1-HaloTag to label the vacuole. Methionine was added for 10–15 min to redistribute Mup1 to PVE compartments, and then cells were subjected to 4D confocal imaging after photobleaching the signal outside the PVE compartments, as described above for the soluble vacuolar cargo. Once again, apparent kiss-and-run transfer was readily observed. The Mup1 cargo moved from PVE compartments to the vacuole, sometimes gradually and sometimes in discrete bursts (*Figure 8—video 1* and *Figure 8A,B*). Rarely, a dramatic event results in transfer of virtually all of the Mup1 in a single burst (*Figure 8—video 2* and *Figure 8C,D*). We never observed complete loss of Vps8 as would be expected for a full fusion event, although a subset of the larger discrete events led to substantial reduction of the Vps8 signal (*Figure 8—video 3* and *Figure 8—figure supplement 1*).

For both the vacuolar cargo and Mup1, quantification revealed that gradual transfer to the vacuole was punctuated, about once every 5 min on average, by bursts in which > 15% of the remaining cargo moved to the vacuole. The average amount of remaining cargo transferred during a burst was ~35–40%. Individual PVE compartments differed greatly in the total fraction of the cargo that was transferred in bursts, but on average this number was ~75%. These results favor a PVE-to-vacuole traffic mechanism involving repeated kiss-and-run events of varying size.

## Discussion

Live-cell fluorescence microscopy complements other methods by providing unique insights into membrane traffic (*Lippincott-Schwartz et al., 2000*). Recently, we engineered a regulatable fluorescent secretory cargo that can be visualized during yeast Golgi maturation (*Casler and Glick, 2019*; *Casler et al., 2019*). This secretory cargo is present in Golgi cisternae throughout the maturation process, and a fraction of the cargo molecules recycle from older to younger cisternae by an AP-1-dependent pathway (*Casler et al., 2019*). As an extension of that work, we have now generated a regulatable fluorescent vacuolar cargo that can be tracked in yeast cells. This vacuolar cargo follows the well characterized pathway by which CPY travels from the Golgi to PVE compartments, with the aid of Vps10 and GGAs, before reaching the vacuole (*Bowers and Stevens, 2005*; *Hecht et al., 2014*).

The convenience of working with an artificial vacuolar cargo is accompanied by some caveats. Compared to CPY, which is monomeric (*Endrizzi et al., 1994*), our vacuolar cargo is a tetramer and thus contains four copies of the Vps10 recognition signal. Moreover, Vps10 has a quality control domain that recognizes the fluorescent protein component of the cargo (*Casler et al., 2019*; *Fitzgerald and Glick, 2014*), so there are multiple points of interaction between Vps10 and the artificial vacuolar cargo. A consequence is that our vacuolar cargo might dissociate from Vps10 more slowly than CPY dissociates, leading to slower PVE-to-vacuole delivery. However, this possible effect on traffic kinetics should not compromise interpretations about the pathway of cargo transport from the Golgi to the vacuole.

Using the vacuolar cargo, we asked a basic question: at what point during cisternal maturation does the vacuolar cargo depart from the Golgi? Unlike the regulatable fluorescent secretory cargo,

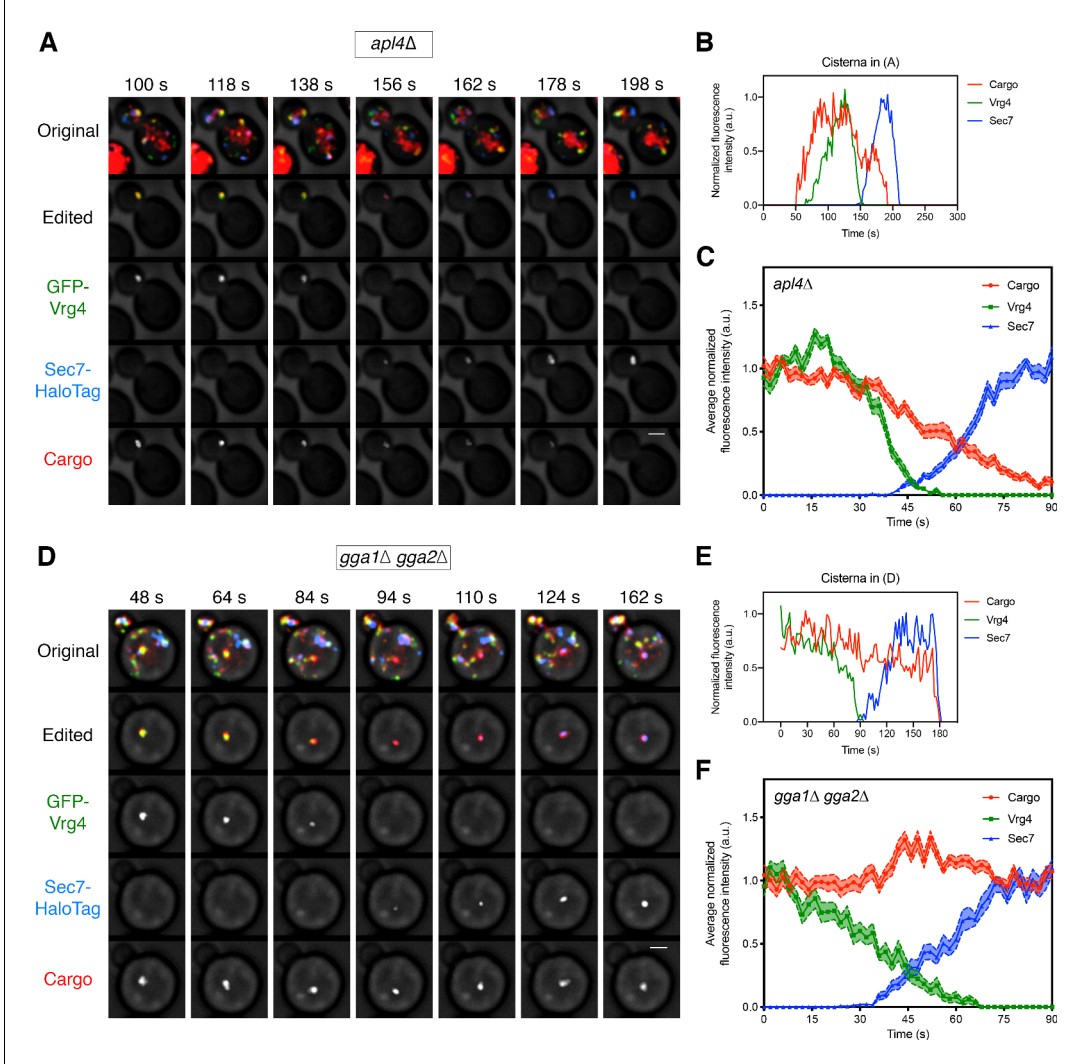

**Figure 6.** Requirement for the GGAs but not AP-1 during Golgi-to-PVE traffic. (**A**) Visualizing vacuolar cargo traffic during Golgi maturation in a strain lacking AP-1. The experiment was performed as in *Figure 4A*, except that an *apl4Δ* strain was used. Shown are average projected Z-stacks at representative time points from *Figure 6—video 1*. The top row shows the complete projection, the second row shows an edited projection that includes only the cisterna being tracked, and the other rows show the individual fluorescence channels from the edited projection. Scale bar, 2 μm. (**B**) Quantification of the fluorescence intensities of the Golgi markers and the vacuolar cargo during a typical maturation event in the *apl4Δ* strain. Depicted are the normalized fluorescence intensities in arbitrary units (a.u.) of the cisterna tracked in (**A**). (**C**) Average cargo signal during the early-to-late Golgi transition in the *apl4Δ* strain. The analysis was performed as in *Figure 4C*, based on 17 maturation events from 13 movies of cells expressing moderate levels of the vacuolar cargo. (**D**) – (**F**) Same as (**A**) – (**C**) except with a *gga1Δ gga2Δ* strain lacking GGAs. The analysis in (**C**) was based on 15 maturation events from 12 movies of cells expressing moderate levels of the vacuolar cargo. Shown in (**D**) are average projected Z-stacks at representative time points from *Figure 6—video 2*.

The online version of this article includes the following video and figure supplement(s) for figure 6:

**Figure supplement 1.** Additional examples of cargo dynamics during cisternal maturation in strains lacking AP-1 or GGAs.

**Figure supplement 2.** Secretion of the vacuolar cargo in cells lacking either Vps10 or GGAs.

**Figure 6—video 1.** Visualizing traffic of the vacuolar cargo during Golgi maturation in a strain lacking AP-1.

https://elifesciences.org/articles/56844#fig6video1

**Figure 6—video 2.** Visualizing traffic of the vacuolar cargo during Golgi maturation in a strain lacking GGAs.

https://elifesciences.org/articles/56844#fig6video2

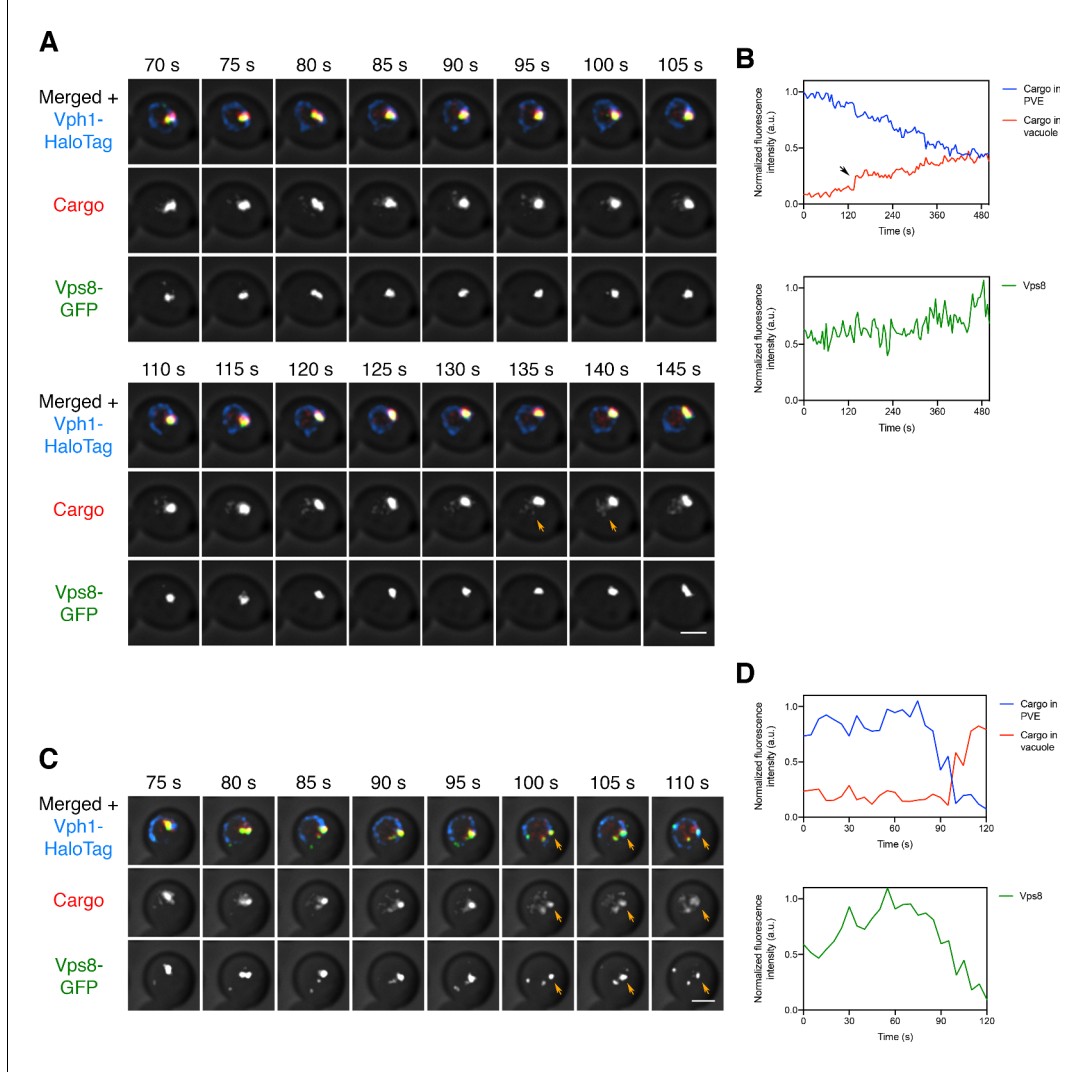

**Figure 7.** Visualizing transfer of the vacuolar cargo from PVE compartments to the vacuole. (**A**) Gradual movement of the vacuolar cargo from a PVE compartment to the vacuole. A strain expressing the vacuolar membrane marker Vph1-HaloTag, the PVE marker Vps8-GFP, and the vacuolar cargo was grown to mid-log phase, attached to a confocal dish, and treated with SLF for 10–15 min to enable the cargo to reach PVE compartments. Prior to imaging, a region that excluded PVE compartments was photobleached by illumination with maximum intensity 561 nm laser light for 40 s. Shown are frames from **Figure 7—video 1**. The top row shows the complete projection, the middle row shows the cargo fluorescence, and the bottom row shows the Vps8-GFP fluorescence. Orange arrows indicate sudden transfer of a small amount of cargo from the PVE compartment to the vacuole. Scale bar, 2 μm. (**B**) Quantification from (**A**) of the time course of cargo fluorescence in the PVE compartment and the vacuole, and of the Vps8 signal. To quantify the cargo signal at each time point, the Vph1 or Vps8 signal was selected in a 3D volume and then the cargo fluorescence within that volume was measured. Normalized data are plotted in arbitrary units (a.u.). The black arrow points to the same cargo transfer event that is marked by the orange arrows in (**A**). (**C**) Example of sudden transfer of a large amount of cargo from a PVE compartment to the vacuole. The experiment was performed as in (**A**). Shown are frames from **Figure 7—video 2**. Orange arrows indicate an event in which nearly all of the cargo moved from the PVE compartment to the vacuole. (**D**) Quantification of (**C**), performed as in (**B**).

The online version of this article includes the following video(s) for figure 7:

**Figure 7—video 1.** Visualizing movement of the vacuolar cargo from a PVE compartment to the vacuole.
https://elifesciences.org/articles/56844#fig7video1

**Figure 7—video 2.** Visualizing sudden movement of the vacuolar cargo from a PVE compartment to the vacuole.
https://elifesciences.org/articles/56844#fig7video2

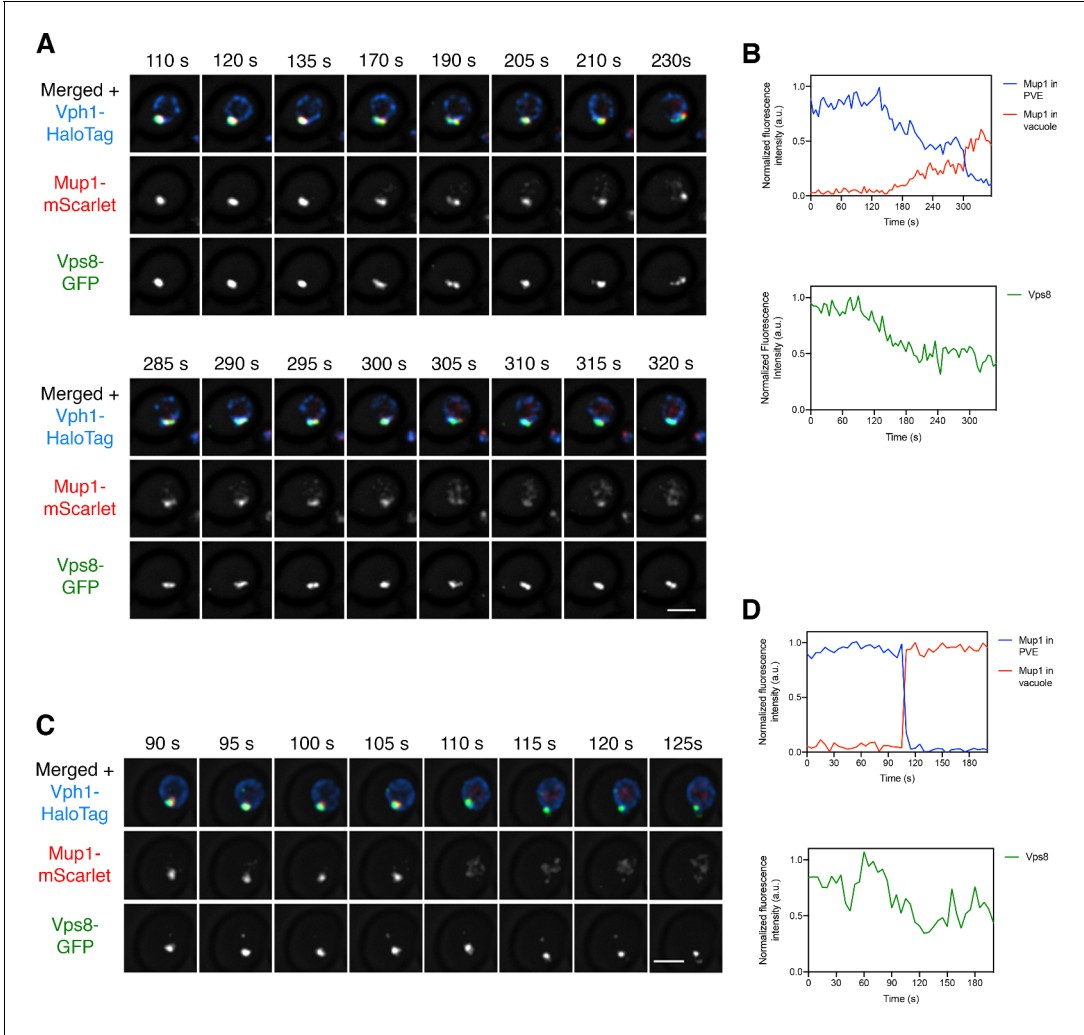

**Figure 8.** Visualizing transfer of Mup1 from PVE compartments to the vacuole. (**A**) Movement of Mup1 from a PVE compartment to the vacuole. A strain expressing the vacuolar membrane marker Vph1-HaloTag, the PVE marker Vps8-GFP, and Mup1-mScarlet was grown to mid-log phase in NSD lacking methionine, attached to a confocal dish, and exposed to NSD containing methionine for 10–15 min to promote internalization of Mup1 to PVE compartments. Prior to imaging, a region that excluded PVE compartments was photobleached by illumination with maximum intensity 561 nm laser light for 5 s. Shown are frames from *Figure 8—video 1*, which illustrates a typical example of putative kiss-and-run fusion at about 300 s. The top row shows the complete projection, the middle row shows the Mup1-mScarlet fluorescence, and the bottom row shows the Vps8-GFP fluorescence. Scale bar, 2 μm. (**B**) Quantification of (**A**), performed as in *Figure 7B*. At about 300 s, a significant amount of Mup1 moved from the PVE compartment to the vacuole. (**C**) Example of an unusually large cargo transfer event. The experiment was performed as in (**A**), and frames are shown from *Figure 8—video 2*. Between the 105 s and 110 s time points, virtually all of the Mup1 moved from the PVE compartment to the vacuole. (**D**) Quantification of (**C**), performed as in *Figure 7B*.

The online version of this article includes the following video and figure supplement(s) for figure 8:

**Figure supplement 1.** Reduction in Vps8 labeling of a PVE compartment after a large cargo transfer event.

**Figure supplement 2.** Evidence from previously published electron tomography data (*McNatt et al., 2007*) for partial fusion of PVE compartments with the vacuole.

**Figure 8—video 1.** Visualizing movement of Mup1 from a PVE compartment to the vacuole.
https://elifesciences.org/articles/56844#fig8video1

**Figure 8—video 2.** Visualizing sudden movement of Mup1 from a PVE compartment to the vacuole.
https://elifesciences.org/articles/56844#fig8video2

**Figure 8—video 3.** Reduction in the apparent size of a PVE compartment after a large cargo transfer event.
https://elifesciences.org/articles/56844#fig8video3

which persists in Golgi cisternae until they are terminally mature (*Casler et al., 2019*), the vacuolar cargo begins to depart during the early-to-late Golgi transition. Departure of the vacuolar cargo begins at about the same time that GGAs arrive at the Golgi, and well before AP-1 arrives. Moreover, departure of the vacuolar cargo is abolished by deleting the GGAs but is unaffected by deleting AP-1. These results fit with previously published data indicating that the CPY pathway from the Golgi to the PVE involves GGAs but not AP-1 (*Dell'Angelica et al., 2000*; *Hirst et al., 2001*; *Hirst et al., 2000*; *Zhdankina et al., 2001*). The new insight is that departure of the vacuolar cargo from the Golgi begins relatively early, about halfway through the time course of cisternal maturation (*Figure 9A*).

Superficially, these results are at odds with the established view that various types of biosynthetic cargoes travel through the entire Golgi until being packaged into distinct carriers at the TGN (*De Matteis and Luini, 2008*; *Griffiths and Simons, 1986*). While there is evidence that biosynthetic cargoes begin to segregate early in the mammalian Golgi, all of those cargoes are thought to reach the terminal TGN compartment (*Chen et al., 2017*). But we suggest that there is actually no discrepancy between the yeast and mammalian data. In a typical cultured mammalian cell, the early Golgi consists of about six cisternae whereas the clathrin-labeled TGN is a single cisterna (*Ladinsky et al., 1999*; *Mogelsvang et al., 2004*). In yeast, the ratio is different because the clathrin-labeled late Golgi/TGN stage occupies about half of the maturation time course (*Figure 9B*). In both cell types, GGA-dependent transport occurs during the TGN stage, but the yeast system has allowed us to define a first TGN sub-stage marked by GGA activity followed by a second TGN sub-stage marked by AP-1 activity (*Figure 9A*). Mammalian TGN structures probably undergo a similar kinetic evolution because the interactions of clathrin adaptors at the mammalian TGN resemble those at the yeast TGN (*Daboussi et al., 2017*). A proposed unified view is that during traffic to either yeast PVE compartments or mammalian late endosomes, GGA-dependent export from the TGN begins prior to secretory vesicle formation and AP-1-dependent intra-Golgi recycling (*Pantazopoulou and Glick, 2019*).

After the vacuolar cargo reaches PVE compartments, how does it move to the vacuole? Previously, based on evidence that PVE compartments are long-lived organelles that rarely if ever fuse completely with the vacuole, we proposed that transfer of material from PVE compartments to the vacuole might involve kiss-and-run fusion events (*Day et al., 2018*). This model seemed plausible because mammalian cells can employ kiss-and-run fusion to exchange material between lysosomes and late endosomes (*Bright et al., 2016*; *Bright et al., 2005*; *Saffi and Botelho, 2019*). Here, we have tested the kiss-and-run hypothesis by fluorescence microscopy. A photobleaching protocol generated a cell in which we could image a single PVE compartment containing fluorescent vacuolar cargo molecules, and the fate of that cohort of molecules was tracked over time. We observed gradual transfer of the cargo molecules to the vacuole over tens of minutes, punctuated by occasional larger events involving sudden transfer of a significant fraction of the cargo molecules. The PVE compartments persisted after cargo transfer. Similar results were seen when PVE compartments were loaded with fluorescently tagged molecules of the methionine permease Mup1, which is internalized for degradation after methionine is added to the medium (*Menant et al., 2006*). Mup1 is packaged into intraluminal vesicles within PVE compartments (*Lin et al., 2008*), so kiss-and-run fusion pores would need to be large enough to transfer intraluminal vesicles. The observed fluctuations in the rate of cargo delivery from PVE compartments to the vacuole suggest that kiss-and-run fusion events vary in size and duration.

Our inference about the kiss-and-run mechanism is based on fluorescence microscopy and will benefit from validation by other techniques. Notably, an electron tomographic reconstruction carried out by the Odorizzi group (*McNatt et al., 2007*), reprinted here as *Figure 8—figure supplement 2*, shows tubular membrane connections between PVE compartments and the vacuole. Our results suggest that such tubular connections should be common, so their prevalence could be quantified in future morphological studies. A separate electron tomographic analysis highlighted a possible complication—PVE compartments were sometimes observed to form clusters (*Adell et al., 2017*). If a single PVE compartment in a cluster fused completely with the vacuole, the fluorescence from the remaining PVE compartments in the cluster would persist, yielding a false impression of kiss-and-run fusion. Although we cannot rule out this scenario, it seems unlikely, because some PVE compartments are presumably solitary and yet we have never seen one disappear by fusing completely with the vacuole. The kiss-and-run hypothesis could be revisited if new methods make it possible either

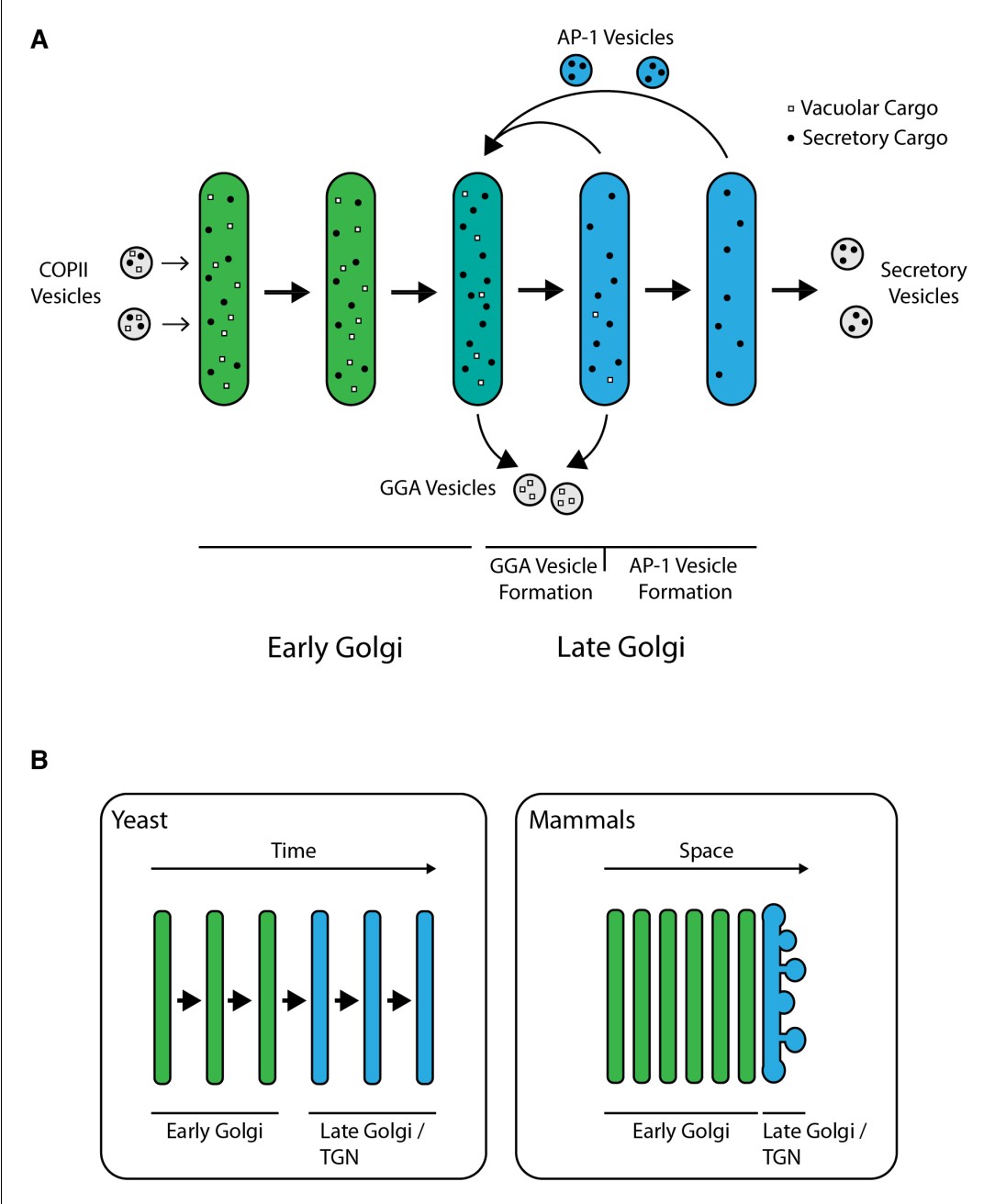

**Figure 9.** Model for sorting of biosynthetic cargoes in the late Golgi. (**A**) Sequential formation of GGA vesicles and AP-1 vesicles in yeast cells. The thick arrows represent progressive maturation of a Golgi cisterna over time. During the early-to-late Golgi transition of cisternal maturation, GGA adaptors arrive, and GGA vesicles that carry vacuolar cargoes (white squares) begin to form. Subsequently, the AP-1 adaptor arrives, and AP-1 vesicles that recycle resident Golgi proteins (not shown) as well as some secretory cargoes (black dots) begin to form. GGAs depart before AP-1 departs, but the formation phases for GGA vesicles and AP-1 vesicles overlap. (**B**) Comparison of Golgi structures in yeast and mammalian cells. In *S. cerevisiae*, the late Golgi or TGN stage accounts for about half of the maturation process, so the sequential arrival times and activities of GGAs and AP-1 are easy to detect. In mammalian cells, Golgi cisternae are stacked, with the youngest early Golgi cisterna at the opposite side of a stack from the oldest late Golgi/TGN cisterna. Only the *trans*-most cisterna of a mammalian Golgi stack functions as late Golgi/TGN, but during the lifetime of this cisterna, GGAs and AP-1 may arrive and act sequentially as in yeast.

to prevent clustering of PVE compartments or to identify solitary PVE compartments by fluorescence microscopy.

If kiss-and-run transfer occurs between PVE compartments and the vacuole, the transient continuities between these organelles might permit exchange of freely diffusing transmembrane proteins. For example, the transmembrane precursor to the vacuolar hydrolase carboxypeptidase S is normally internalized into intraluminal vesicles of PVE compartments, but a mutant carboxypeptidase S precursor that cannot be internalized moves from the limiting membranes of PVE compartments to the vacuole membrane (*Katzmann et al., 2001*). Although other transmembrane proteins such as Vps10 and the SNARE protein Pep12 normally reside in PVE compartments and not in the vacuole membrane, overexpressed Vps10 and Pep12 do reach the vacuole membrane (*Black and Pelham, 2000*; *Cereghino et al., 1995*; *Chi et al., 2014*). It seems plausible that during transient fusions of PVE compartments with the vacuole, saturable interactions with the trafficking machinery retain proteins such as Vps10 and Pep12 in the PVE compartments. Hence, the kiss-and-run model can explain how some transmembrane PVE proteins reach the vacuole while others do not, whereas this phenomenon would be hard to understand if PVE compartments fused completely with the vacuole.

Cargo transfer from PVE compartments to the vacuole is slow relative to other membrane traffic steps in budding yeast. Historically, kinetics of traffic to the vacuole have been measured by pulse-chase experiments that rely on the maturation of CPY by vacuolar proteases (*Stevens et al., 1982*; *Vida et al., 1993*), but those results may be misleading because kiss-and-run fusion could result in some vacuolar proteases being present and active in PVE compartments (*Bright et al., 2016*). As judged by fluorescence microscopy, PVE-to-vacuole transfer often requires 30 min or more to reach completion. By comparison, transit from the ER to the PVE along the biosynthetic pathway or from the plasma membrane to the PVE along the endocytic pathway is largely complete within 10 min (*Casler et al., 2019*; *Day et al., 2018*; *Losev et al., 2006*). A possible reason for PVE-to-vacuole traffic being rate-limiting is that the cell has the option to 'change its mind' by recycling membrane proteins from the PVE instead of degrading them (*Ma and Burd, 2020*). Meanwhile, the AP-3-driven pathway of direct Golgi-to-vacuole traffic offers an alternative for rapid delivery of membrane proteins to the vacuole (*Odorizzi et al., 1998*).

# Materials and methods

## Key resources table

| Reagent type (species) or resource | Designation | Source or reference | Identifiers | Additional information |
|---|---|---|---|---|
| Chemical compound, drug | Hygromycin | Thermo Fisher | Cat. #: 10687010 | |
| Chemical compound, drug | G418 | Teknova | Cat. #: G5001 | |
| Chemical compound, drug | Nourseothricin | Neta Scientific | Cat. #: RPI-N51200-1.0 | |
| Chemical compound, drug | JF$_{646}$ HaloTag Ligand | Dr. Luke Lavis (Janelia Research Campus) *Grimm et al., 2015* | | |
| Chemical compound, drug | SLF | Cayman Chemical | Cat. #: 10007974–5 | |
| Chemical compound, drug | Cycloheximide | Neta Scientific | Cat. #: RPI-C81040-1.0 | |
| Chemical compound, drug | Concanavalin A | Sigma-Aldrich | Cat. #: C2010-250MG | |
| Antibody | anti-FKBP12 (rabbit polyclonal) | Abcam | Cat. #: ab2918 | WB (1:1000) |
| Antibody | Alexa Fluor 647 anti-rabbit (goat polyclonal) | Thermo Fisher | Cat. #: A21245 | WB (1:1000) |

*Continued on next page*

*Continued*

| Reagent type (species) or resource | Designation | Source or reference | Identifiers | Additional information |
|---|---|---|---|---|
| Software, algorithm | Graphpad Prism | Insightful Science (https://www.graphpad.com) | RRID:SCR_002798 | |
| Software, algorithm | SnapGene | Insightful Science (https://www.snapgene.com) | RRID:SCR_015052 | |
| Software, algorithm | ImageJ | ImageJ (https://imagej.nih.gov/ij/) | RRID:SCR_003070 | |

## Yeast growth and strain construction

The parental haploid strain was JK9-3da (*leu2-3,112 ura3-52 rme1 trp1 his4*) (*Kunz et al., 1993*). Yeast were grown with shaking in baffled flasks at 23°C in nonfluorescent minimal glucose dropout medium (NSD) (*Bevis et al., 2002*) or in rich glucose medium (YPD) supplemented with adenine and uracil.

Deletion of the *PDR1*, *PDR3*, and *GGA1* genes was accomplished by replacement with a G418, nourseothricin, or hygromycin resistance cassette from pFA6a-kanMX6, pAG25, or pAG32, respectively (*Goldstein and McCusker, 1999*; *Wach et al., 1994*). Deletion of *VPS10*, *APM3*, and *APL4* was accomplished by using overlap extension PCR to generate a hygromycin resistance cassette, amplified from pAG32, flanked by 500 bp upstream and downstream of the gene. Deletion of *GGA2* was accomplished in the same manner, except that the *LEU2* gene from *K. lactis* was amplified from pUG73 (*Gueldener et al., 2002*). The primers used for these procedures are listed in *Table 1*.

Yeast proteins were tagged by gene replacement using the pop-in/pop-out method to maintain endogenous expression levels (*Rossanese et al., 1999*; *Rothstein, 1991*). Secretory and vacuolar cargo proteins were expressed using a *TRP1* integrating vector with the strong constitutive *TPI1* promoter and the *CYC1* terminator (*Fitzgerald and Glick, 2014*). To ensure consistent expression levels, each strain was verified to have a single copy of the integrated plasmid by PCR with the primers listed in *Table 1*. All plasmids used in this study are documented in the online supplemental material ZIP file, which contains annotated map/sequence files that can be opened with SnapGene Viewer (Insightful Science; https://www.snapgene.com/snapgene-viewer/). Newly generated plasmids have been archived with Addgene (https://www.addgene.org/Benjamin_Glick/, catalog numbers 140149, 140150, 140151, 140152, 140153, 140154).

## Fluorescence microscopy and photobleaching

For live-cell fluorescence imaging, yeast strains were grown in NSD (pH ~5.5) at 23°C. Where indicated, SLF was diluted from a 100 mM stock solution in ethanol (Cayman Chemical; 10007974) to a final concentration of 100 µM, and cycloheximide was added from a 100 mg/mL stock solution in DMSO. Cells were attached to a concanavalin A-coated coverglass-bottom dish containing NSD (*Losev et al., 2006*) for imaging on a Leica SP8 or Leica SP5 confocal microscope equipped with a 1.4 NA/63x oil objective, using a pixel size of 60–80 nm, a Z-step interval of 0.25–0.30 µm, and 20–30 optical sections. The intervals between Z-stacks were based on the requirements for the individual experiments.

Static images and 4D movies were processed as follows. Static images were converted to 16-bit and average projected (*Hammond and Glick, 2000*), then range-adjusted to the minimum and maximum pixel values with ImageJ (*Schneider et al., 2012*). Movies were deconvolved with Huygens Essential (Scientific Volume Imaging) using the classic maximum likelihood estimation algorithm (*Day et al., 2017*). Movies were converted to hyperstacks and average projected, then range-adjusted to maximize contrast in ImageJ. Custom ImageJ plugins were used to generate montages of time series, select individual structures and remove extraneous structures, convert edited montages to hyperstacks, and measure fluorescence intensities (*Johnson and Glick, 2019*). Each kinetic trace of a fluorescent Golgi marker was normalized to the average of the three highest values measured, a method that provided better results than relying on the single highest signal from a noisy data set.

**Table 1.** Primers used in this study.

| Purpose | Amplifies | Primers |
|---|---|---|
| *PDR1* deletion | kanMX resistance cassette | 5'-CAGCCAAGAATATACAGAAAAGAATCCAAGAAACTGGAAGCGTACGCTGCAGGTCGAC-3' 5'-GGAAGTTTTTGAGAACTTTTATCTATACAAACGTATACGTATCGATGAATTCGAGCTCG-3' |
| *PDR3* deletion | Nourseothricin resistance cassette | 5'-ATCAGCAGTTTTATTAATTTTTTCTTATTGCGTGACCGCACGTACGCTGCAGGTCGAC-3' 5'-TACTATGGTTATGCTCTGCTTCCCTATTTTCTTTGCGTTTATCGATGAATTCGAGCTCG-3' |
| *GGA1* deletion | Hygromycin resistance cassette | 5'-AGTCACTACTTCAAGTATAACCCAGACAAGAGTCTTTTAAATAGCTTGCCTTGTCCCCGC-3' 5'-ATGGCATCTACTTTTTTTTTCAACTTCTCTACCGAATTTGACGTTTTCGACACTGGATGGC-3' |
| *VPS10* deletion | *VPS10* 5' upstream | 5'-CCCAAACTAAAAAGTATCCGCCTGT-3' 5'-GACAAGGCAAGCTAACGTGTGATGACTACTGGACACT-3' |
| | *VPS10* 3' downstream | 5'-GCCATCCAGTGTCGAAGAGATTACTTTACATAGAGTAGATAATTCCATATACTTTTCATA –3' 5'-AATGAAGTACTATAAATATTAAAGTACGTTAGTAGTTTATTTCTCTTCGG-3' |
| | Hygromycin resistance cassette | 5'-TCATCACACGTTAGCTTGCCTTGTCCCCGC-3' 5'-TGTAAAGTAATCTCTTCGACACTGGATGGCGG-3' |
| *APM3* deletion | *APM3* 5' upstream | 5'-AGGGGTAGAAGTCGCTGATTGAT-3' 5'-GGGCCTCCATGTCCTATTTTGGTTGGGTTGGTAAGGTTTACAG-3' |
| | *APM3* 3' downstream | 5'-GCTGGTCGCTATACTGTTATATGTGTACTTGAAATTCCATGCGAAACTAAA-3' 5'-TGCGGAAGTCTTCCCTAAGACG-3' |
| | Hygromycin resistance cassette | 5'-CAACCAAAATAGGACATGGAGGCCCAGAATACCC-3' 5'-TCAAGTACACATATAACAGTATAGCGACCAGCATTCACA-3' |
| *APL4* deletion | *APL4* 5' upstream | 5'-ATGTATATAATTCCGGAAGTGTGGTCCT-3' 5'-GACAAGGCAAGCTTATGGTGTTCAGGTCTTTCTCGTTGCT-3' |
| | *APL4* 3' downstream | 5'-CCATCCAGTGTCGAAAAATGCCTTTAAAATTACAGAACATAACATGATTAATGAC-3' 5'-GAATTCTGGTCCAAGGCAATTCTATATTTGAT-3' |
| | Hygromycin resistance cassette | 5'-CCTGAACACCATAAGCTTGCCTTGTCCCCG-3' 5'-TTTTAAAGGCATTTTTCGACACTGGATGGCGG-3' |

*Table 1 continued on next page*

*Table 1 continued*

| Purpose | Amplifies | Primers |
|---|---|---|
| GGA2 deletion | GGA2 5' upstream | 5'-GATTTCTACAGTCTTTC TGATGGGTTCTTGG-3' 5'-ACGATATTCTTAGACATGAT GCAGTATCACGATTAGCAAT-3' |
| | GGA2 3' downstream | 5'-AATCTTGGCTTAATCCTCTG GCGTTTCTTATCAATCCTTTCT-3' 5'-TCTTCCTTTGAAGAAAA TTCGTCCTCATCT-3' |
| | K. lactis LEU2 | 5'-AATCGTGATACTGCATCATG TCTAAGAATATCGTTGTCCTACCGG-3' 5'-GAAACGCCAGAGGATTAAG CCAAGATTTCCTTGACAGCC-3' |
| Integration at TRP1 | TRP1 locus | 5'-GTGTACTTTGCAGTTATGACG-3' 5'-AGTCAACCCCCTGCGATGTATATTTTCCTG-3' |

For photobleaching prior to 4D imaging of PVE compartments, a region of interest was drawn to include all fluorescent structures within the cell except the PVE compartments. The regulable vacuolar cargo and Mup1-mScarlet were bleached by maximum-intensity illumination with a 561 nm laser for 40 s or 5 s, respectively. These bleaching durations were chosen by determining the minimal times needed to bleach the fluorescence signals completely.

To quantify the frequency at which bursts of cargo moved from PVE compartments to the vacuole, each 10 min video was examined at 10 s intervals. Bursts were scored as intervals in which more than 15% of the remaining cargo transferred to the vacuole. Six movies for the vacuolar cargo and seven movies for Mup1 were analyzed. The two cargoes displayed similar frequencies of large bursts as well as similar amounts of cargo transferred per burst, so the data were combined to obtain the averaged numbers stated in the text.

## HaloTag labeling

Proteins modified with HaloTag were labeled as previously described (*Casler et al., 2019*). Briefly, 1 µL of a 1 mM stock solution of JF$_{646}$ ligand (*Grimm et al., 2015*) in DMSO was diluted in 500 µL NSD and then spun at maximum speed in a microcentrifuge to remove precipitated material, and the supernatant was mixed with a 500 µL aliquot of cells. Labeling was performed for 30 min with shaking at 23°C. To remove excess dye, cells were filter-washed by pushing 3 mL of fresh medium through the filter. Washed cells were resuspended by pipetting on the filter. The resulting cell mixture was diluted to its original density, and the cells were attached to confocal dishes.

## Immunoblotting of yeast cell lysates and secreted proteins

Immunoblotting was performed as follows *Casler et al., 2019*. A 5 mL yeast culture was grown in YPD overnight with shaking in a baffled flask to an OD$_{600}$ of 0.7–1.0. The cells were collected by a brief spin in a microcentrifuge, washed twice with fresh YPD, and resuspended in the original volume of fresh YPD. Cultures were treated with 100 µM SLF, and 1.6 mL was removed at each time point. The cells were collected by spinning at 2500xg (5000 rpm) for 2 min in a microcentrifuge. The culture medium supernatant was transferred to a fresh microcentrifuge tube on ice, and the cells were washed once with deionized water. Then the cells were resuspended in 100 µL 4% trichloroacetic acid. Glass beads (0.5 mm; BioSpec Products) were added to bring the total volume to ~200 µL, and the sample was vortexed three times for 1 min each separated by 1 min intervals on ice. Finally, 800 µL of PBS was added, the solution was mixed, and 800 µL of the cell lysate was transferred to a fresh microcentrifuge tube. Supernatant protein samples were precipitated with 4% trichloroacetic acid on ice for 20 min. Precipitated proteins were centrifuged at maximum speed in a microcentrifuge for 15 min at 4°C. Finally, each protein pellet was resuspended in 50 µL SDS-PAGE sample buffer.

Treatment with endoglycosidase H was performed as described by the manufacturer (New England Biolabs; P0702S). Briefly, Glycoprotein Denaturing Buffer was added to the protein sample,

which was boiled for 5–10 min, followed by addition of GlycoBuffer 3 and endoglycosidase H. The reaction was performed at 37°C for at least 1 hr.

For immunoblotting, 9 µL of each cell lysate and 14 µL of each secreted protein sample were run on a 4–20% Tris-glycine gel (Bio-Rad; 4561094) together with the Precision Plus Protein Dual Color Standards molecular weight marker (Bio-Rad; 1610374). The separated proteins were transferred to a PVDF membrane (Bio-Rad; 1704156) using the Trans-Blot Turbo system (Bio-Rad). The membrane was blocked with 5% nonfat dry milk in TBST (50 mM Tris-HCl at pH 7.6, 150 mM NaCl, and 0.05% Tween 20) with shaking at room temperature for 1 hr, and then incubated with 1:1000 polyclonal rabbit anti-FKBP12 antibody (Abcam; ab2918) in 5% milk/TBST with shaking overnight at 4°C. After four 5 min washes in TBST, the membrane was incubated with a 1:1000 dilution of goat anti-rabbit secondary antibody conjugated to Alexa Fluor 647 (Thermo Fisher Scientific; A21245) for 1 hr at room temperature. The membrane was then washed twice with TBST and twice with 50 mM Tris-HCl at pH 7.6, 150 mM NaCl. Analysis was performed with a LI-COR Odyssey CLx imaging system.

## Acknowledgements

Thanks for assistance with fluorescence microscopy to Christine Labno and Vytas Bindokas at the Integrated Microscopy Core Facility. Additional thanks to Matt West and Greg Odorizzi for alerting us to their relevant electron tomography data and for providing an image to be reproduced here, to Luke Lavis for providing the $JF_{646}$ dye, and to members of the Glick lab for helpful feedback.

This work was supported by National Institutes of Health (NIH) grant R01 GM104010. JC Casler was supported by NIH training grant T32 GM007183. The Integrated Microscopy Core Facility is supported by the NIH-funded Cancer Center Support Grant P30 CA014599.

## Additional information

### Funding

| Funder | Grant reference number | Author |
| --- | --- | --- |
| National Institutes of Health | R01 GM104010 | Benjamin S Glick |
| National Institutes of Health | T32 GM007183 | Jason C Casler |
| National Institutes of Health | P30 CA014599 | Benjamin S Glick |

The funders had no role in study design, data collection and interpretation, or the decision to submit the work for publication.

### Author contributions

Jason C Casler, Conceptualization, Formal analysis, Investigation, Methodology, Writing - original draft; Benjamin S Glick, Conceptualization, Resources, Supervision, Funding acquisition, Writing - review and editing

### Author ORCIDs

Jason C Casler http://orcid.org/0000-0001-9742-9978
Benjamin S Glick https://orcid.org/0000-0002-7921-1374

### Decision letter and Author response

Decision letter https://doi.org/10.7554/eLife.56844.sa1
Author response https://doi.org/10.7554/eLife.56844.sa2

## Additional files

### Supplementary files

- Supplementary file 1. Plasmid files.
- Transparent reporting form

## Data availability

Newly created plasmids have been archived with Addgene (https://www.addgene.org/Benjamin_Glick/, catalog numbers 140149, 140150, 140151, 140152, 140153, 140154). Yeast strains are freely available upon request to any interested researcher.

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
