## [Decision Letter]

**Acceptance summary:**

This study provides compelling evidence that a yeast vacuolar precursor protein is sorted from an earlier Golgi compartment than previously thought, and secondly, that transfer of vacuolar cargo from the pre-vacuolar endosome to the vacuole is gradual and does not 'consume' the endosome. The results suggest that cargo transfer between the pre-vacuolar endosome and the vacuole occurs via a 'kiss and run' mechanism.

**Decision letter after peer review:**

Thank you for submitting your article "A microscopy-based kinetic analysis of yeast vacuolar protein sorting" for consideration by *eLife*. Your article has been reviewed by three peer reviewers, including Christopher G Burd as the Reviewing Editor and reviewer #1, and the evaluation has been overseen by Vivek Malhotra as the Senior Editor. The following individual involved in review of your submission has agreed to reveal their identity: David Teis (Reviewer #2).

The reviewers have discussed the reviews with one another and the Reviewing Editor has drafted this decision to help you prepare a revised submission.

Summary:

The reviewers found your manuscript to clearly present a well-executed study of biosynthetic and endocytic protein trafficking to the yeast vacuole. Regarding the two major conclusions of the manuscript, all three reviewers considered the finding that a fluorescent vacuolar enzyme reporter is exported from the Golgi at the early-to-late cisterna transition interesting and well supported by the analyses. The reviewers found the analysis of content transfer between the pre-vacuolar endosome and the vacuole interesting and potentially important. However, each reviewer considered the conclusion that content is transferred by a kiss and run fusion mechanism to be poorly supported. Their consensus opinion is that substantial additional experimentation is necessary to rigorously prove this point. The core points that the reviewers suggest need to be addressed are the following.

1) The use of Vps8 as an accurate marker of the PVE needs to be more firmly established. It is a component of the CORVET complex that tethers clusters of late endosomes with Vps21, hence, it is a reasonable possibility that individual late endosomes from the cluster fuse with the vacuole. The reviewers also point several concerns with tagged/inappropriately expressed Vps8 that need to be addressed.

2) An ultra-structural analysis of the PVE and vacuole is required to clarify the nature of the tethered PVE (single vs. multiple compartments) and to define the structural relationship between the PVE and vacuole (fused?). The single reprinted micrograph showing MVBs apparently fused to the vacuole membrane via a stalk underscores the importance of the approach, which seems suitable for rigorously characterizing the PVE-vacuole junctions.

Given the impact of the COVID-19 pandemic to the research community, the reviewers considered the strength of the study as it currently stands. From this perspective, the reviewers agreed that the results presented in the study would provide a substantial and healthy advance to the field if the following considerations were addressed by additional analysis of data already in hand and modifications to the presentation, as follows:

1) The presentation of transfer by kiss and run fusion should be consistent throughout the manuscript. For example, the Abstract states that the data "imply" kiss and run. The last sentence of the Introduction states that transfer during kiss and run fusion has been "demonstrate(d)". The reviewers agree that the results can be explained by a kiss and run fusion mechanism, as indicated in the Abstract, but not that cargo transfer via kiss and run fusion has been demonstrated. The text should accurately and consistently indicate that kiss and run fusion remains to be demonstrated.

2) In Figures 7B/C and 8B/C, the two distinct types of PVE-to-vacuole transfer events (slow/gradual versus rapid) should be quantified so that it can be determined which occurrence, if either, is most prevalent.

Revisions expected in follow-up work:

An ultra-structural analysis of the PVE and vacuole is required to clarify the nature of the tethered PVE (single vs. multiple compartments) and to define the spatial relationship between the PVE and vacuole.

---

## [Author Response]

Summary:The reviewers found your manuscript to clearly present a well-executed study of biosynthetic and endocytic protein trafficking to the yeast vacuole. […] The core points that the reviewers suggest need to be addressed are the following.1) The use of Vps8 as an accurate marker of the PVE needs to be more firmly established. It is a component of the CORVET complex that tethers clusters of late endosomes with Vps21, hence, it is a reasonable possibility that individual late endosomes from the cluster fuse with the vacuole. The reviewers also point several concerns with tagged/inappropriately expressed Vps8 that need to be addressed.

If we understand correctly, the concern is that Vps8 might specifically mark clusters of PVE compartments and not label individual PVE compartments. We have enhanced the text to allay this concern. In our previous work (Day et al., 2018), we showed that Vps8GFP colocalizes with a variety of other PVE markers, and that PVE compartments showed similar dynamics when labeled with Vps8-GFP or with other PVE markers. There was no hint that any PVE compartments might fail to label with Vps8.

The use of Vps8-GFP as a PVE marker is based on a paper from the Ungermann lab (Arlt et al., 2015), who have studied Vps8 and related topics extensively.

2) An ultra-structural analysis of the PVE and vacuole is required to clarify the nature of the tethered PVE (single vs. multiple compartments) and to define the structural relationship between the PVE and vacuole (fused?). The single reprinted micrograph showing MVBs apparently fused to the vacuole membrane via a stalk underscores the importance of the approach, which seems suitable for rigorously characterizing the PVE-vacuole junctions.

We agree that ultrastructural analysis would be needed for a definitive conclusion about the mechanism of cargo transfer from PVE compartments to the vacuole. The reprinted micrograph is included here because it suggests that our interpretation is plausible. We have modified the text to be appropriately cautious.

Revisions for this paper:Given the impact of the COVID-19 pandemic to the research community, the reviewers considered the strength of the study as it currently stands. From this perspective, the reviewers agreed that the results presented in the study would provide a substantial and healthy advance to the field if the following considerations were addressed by additional analysis of data already in hand and modifications to the presentation, as follows:1) The presentation of transfer by kiss and run fusion should be consistent throughout the manuscript. For example, the Abstract states that the data "imply" kiss and run. The last sentence of the Introduction states that transfer during kiss and run fusion has been "demonstrate(d)". The reviewers agree that the results can be explained by a kiss and run fusion mechanism, as indicated in the Abstract, but not that cargo transfer via kiss and run fusion has been demonstrated. The text should accurately and consistently indicate that kiss and run fusion remains to be demonstrated.

We fully agree. The text has been modified to emphasize that kiss-and-run is merely a hypothesized mechanism that can potentially explain the data, and that this hypothesis needs to be tested by future analysis.

2) In Figures 7B/C and 8B/C, the two distinct types of PVE-to-vacuole transfer events (slow/gradual versus rapid) should be quantified so that it can be determined which occurrence, if either, is most prevalent.

This quantification is now included in the text at the end of the Results section. The procedure is described in a new paragraph in the Materials and methods section.

Revisions expected in follow-up work:An ultra-structural analysis of the PVE and vacuole is required to clarify the nature of the tethered PVE (single vs. multiple compartments) and to define the spatial relationship between the PVE and vacuole.

We acknowledge the limitations of the current data, but further thought will be needed for a suitable follow-up analysis. An ideal experiment would reveal ultrastructural data of the same PVE compartments before and after bursts of cargo transfer to the vacuole, but such an experiment is not feasible. We are more optimistic about the prospect of distinguishing single from clustered PVE compartments by fluorescence microscopy, perhaps with some form of super-resolution imaging. However, such a method remains to be developed.